# TTN: A Domain-Shift Aware Batch Normalization in Test-Time Adaptation

**Hyesu Lim**[1,2]*, **Byeonggeun Kim**\*, **Jaegul Choo**[2], **Sungha Choi**[1]‡
[1]Qualcomm AI Research†,  [2]KAIST

## Abstract

This paper proposes a novel batch normalization strategy for test-time adaptation. Recent test-time adaptation methods heavily rely on the modified batch normalization, *i.e.,* transductive batch normalization (TBN), which calculates the mean and the variance from the current test batch rather than using the running mean and variance obtained from source data, *i.e.,* conventional batch normalization (CBN). Adopting TBN that employs test batch statistics mitigates the performance degradation caused by the domain shift. However, re-estimating normalization statistics using test data depends on impractical assumptions that a test batch should be large enough and be drawn from i.i.d. stream, and we observed that the previous methods with TBN show critical performance drop without the assumptions. In this paper, we identify that CBN and TBN are in a trade-off relationship and present a new *test-time normalization* (TTN) method that interpolates the standardization statistics by adjusting the importance between CBN and TBN according to the *domain-shift sensitivity* of each BN layer. Our proposed TTN improves model robustness to shifted domains across a wide range of batch sizes and in various realistic evaluation scenarios. TTN is widely applicable to other test-time adaptation methods that rely on updating model parameters via backpropagation. We demonstrate that adopting TTN further improves their performance and achieves state-of-the-art performance in various standard benchmarks.

## 1 Introduction

When we deploy deep neural networks (DNNs) trained on the source domain into test environments (*i.e.,* target domains), the model performance on the target domain deteriorates due to the domain shift from the source domain. For instance, in autonomous driving, a well-trained DNNs model may exhibit significant performance degradation at test time due to environmental changes, such as camera sensors, weather, and region (Choi et al., 2021; Lee et al., 2022; Kim et al., 2022b).

Test-time adaptation (TTA) has emerged to tackle the distribution shift between source and target domains during test time (Sun et al., 2020; Wang et al., 2020). Recent TTA approaches (Wang et al., 2020; Choi et al., 2022; Liu et al., 2021) address this issue by 1) (re-)estimating normalization statistics from current test input and 2) optimizing model parameters in unsupervised manner, such as entropy minimization (Grandvalet & Bengio, 2004; Long et al., 2016; Vu et al., 2019) and self-supervised losses (Sun et al., 2020; Liu et al., 2021). In particular, the former focused on the weakness of conventional batch normalization (CBN) (Ioffe & Szegedy, 2015) for domain shift in a test time. As described in Fig. 1(b), when standardizing target feature activations using source statistics, which are collected from the training data, the activations can be transformed into an unintended feature space, resulting in misclassification. To this end, the TTA approaches (Wang et al., 2020; Choi et al., 2022; Wang et al., 2022) have heavily depended on the direct use of test batch statistics to fix such an invalid transformation in BN layers, called transductive BN (TBN) (Nado et al., 2020; Schneider et al., 2020; Bronskill et al., 2020) (see Fig. 1(c)).

The approaches utilizing TBN showed promising results but have mainly been assessed in limited evaluation settings (Wang et al., 2020; Choi et al., 2022; Liu et al., 2021). For instance, such evaluation settings assume large test batch sizes (*e.g.,* 200 or more) and a single stationary distribution shift

---

*Work completed while at Qualcomm Technologies, Inc.   ‡Corresponding author.
†Qualcomm AI Research is an initiative of Qualcomm Technologies, Inc.

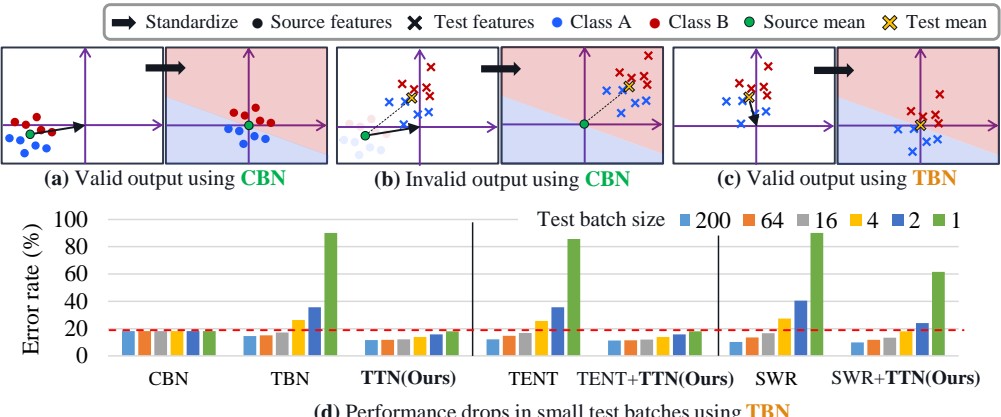

**(a)** Valid output using **CBN**     **(b)** Invalid output using **CBN**     **(c)** Valid output using **TBN**

**(d)** Performance drops in small test batches using **TBN**

Figure 1: **Trade-off between CBN & TBN.** In conceptual illustrations (a), (b), and (c), the depicted standardization only considers making the feature distribution have a zero mean, disregarding making it have unit variance. When the source and test distributions are different, and the test batch size is large, **(b)** test features can be wrongly standardized when using CBN (Ioffe & Szegedy, 2015), but **(c)** TBN (Nado et al., 2020) can provide a valid output. **(d)** Error rates ($\downarrow$) on shifted domains (CIFAR-10-C). TBN and TBN applied (TENT (Wang et al., 2020), SWR (Choi et al., 2022)) methods suffer from severe performance drop when the batch size becomes small, while TTN (Ours) improves overall performance.

(*i.e.,* single corruption). Recent studies suggest more practical evaluation scenarios based on small batch sizes (Mirza et al., 2022; Hu et al., 2021; Khurana et al., 2021) or continuously changing data distribution during test time (Wang et al., 2022). We show that the performance of existing methods significantly drops once their impractical assumptions of the evaluation settings are violated. For example, as shown in Fig. 1(d), TBN (Nado et al., 2020) and TBN applied methods suffer from severe performance drop when the test batch size becomes small, while CBN is irrelevant to the test batch sizes. We identify that CBN and TBN are in a trade-off relationship (Fig. 1), in the sense that one of each shows its strength when the other falls apart.

To tackle this problem, we present a novel *test-time normalization (TTN)* strategy that controls the trade-off between CBN and TBN by adjusting the importance of source and test batch statistics according to the *domain-shift sensitivity* of each BN layer. Intuitively, we linearly interpolate between CBN and TBN so that TBN has a larger weight than CBN if the standardization needs to be adapted toward the test data. We optimize the interpolating weight after the pre-training but before the test time, which we refer to as the post-training phase. Specifically, given a pre-trained model, we first estimate channel-wise sensitivity of the affine parameters in BN layers to domain shift by analyzing the gradients from the back-propagation of two input images, clean input and its augmented one (simulating unseen distribution). Afterward, we optimize the interpolating weight using the channel-wise sensitivity replacing BN with the TTN layers. It is noteworthy that none of the pre-trained model weights are modified, but we only train newly added interpolating weight.

We empirically show that TTN outperforms existing TTA methods in realistic evaluation settings, *i.e.,* with a wide range of test batch sizes for single, mixed, and continuously changing domain adaptation through extensive experiments on image classification and semantic segmentation tasks. TTN as a stand-alone method shows compatible results with the state-of-the-art methods and combining our TTN with the baselines even boosts their performance in overall scenarios. Moreover, TTN applied methods flexibly adapt to new target domains while sufficiently preserving the source knowledge. No action other than computing per batch statistics (which can be done simultaneously to the inference) is needed in test-time; TTN is compatible with other TTA methods without requiring additional computation cost.

Our contributions are summarized as follows:

- We propose a novel domain-shift aware test-time normalization (TTN) layer that combines source and test batch statistics using channel-wise interpolating weights considering the sensitivity to domain shift in order to flexibly adapt to new target domains while preserving the well-trained source knowledge.

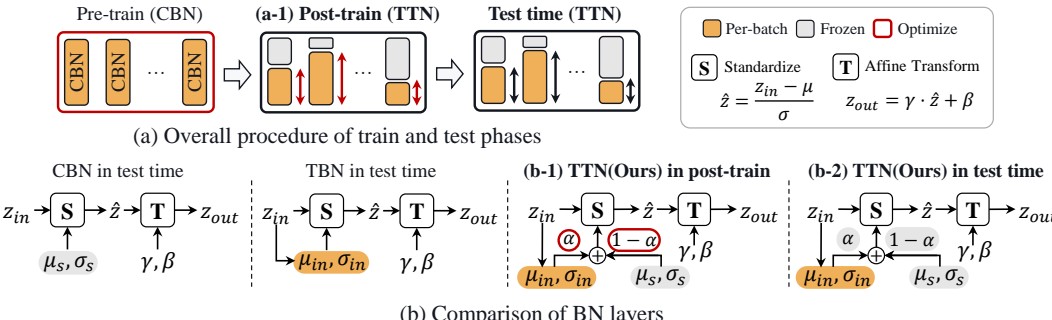

Figure 2: **Method overview. (a)** We introduce an additional training phase between pre-train and test time called **(a-1)** post-training phase. **(b)** Our proposed TTN layer combines per-batch statistics and frozen source statistics with interpolating weight $\alpha$, which is **(b-1)** optimized in post-training phase and **(b-2)** fixed in test time.

- To show the broad applicability of our proposed TTN, which does not alter training or test-time schemes, we show that adding TTN to existing TTA methods significantly improves the performance across a wide range of test batch sizes (from 200 to 1) and in three realistic evaluation scenarios; stationary, continuously changing, and mixed domain adaptation.

- We evaluate our method through extensive experiments on image classification using CIFAR-10/100-C, and ImageNet-C (Hendrycks & Dietterich, 2018) and semantic segmentation task using CityScapes (Cordts et al., 2016), BDD-100K (Yu et al., 2020), Mapillary (Neuhold et al., 2017), GTAV (Richter et al., 2016), and SYNTHIA (Ros et al., 2016).

## 2 METHODOLOGY

In this section, we describe our method, the *Test-Time Normalization* (TTN) layer, whose design is suitable for test-time adaptation (TTA) in practical usages out of the large batch size and i.i.d assumptions during a test time. We first define the problem setup in Section 2.1 and present our proposed TTN layers in Section 2.2. Finally, we discuss how we optimize TTN layers in Sections 2.3.

### 2.1 PROBLEM SETUP

Let the train and test data be $\mathcal{D}_S$ and $\mathcal{D}_T$ and the corresponding probability distributions be $P_S$ and $P_T$, respectively, where $\mathcal{D}_S$ and $\mathcal{D}_T$ share the output space, *i.e.*, $\{y_i\} \sim \mathcal{D}_S = \{y_i\} \sim \mathcal{D}_T$. The covariate shift in TTA is defined as $P_S(x) \neq P_T(x)$ where $P_S(y|x) = P_T(y|x)$ (Quinonero-Candela et al., 2008). A model, $f_\theta$, with parameters $\theta$, is trained with a mini-batch, $\mathcal{B}^S = \{(x_i, y_i)\}_{i=1}^{|\mathcal{B}^S|}$, from source data $\mathcal{D}_S$, where $x_i$ is an example and $y_i$ is the corresponding label. During the test, $f_\theta$ encounters a test batch $\mathcal{B}^T \sim \mathcal{D}_T$, and the objective of TTA is correctly managing the test batch from the different distribution.

To simulate more practical TTA, we mainly consider two modifications: (1) various test batch sizes, $|\mathcal{B}^T|$, where small batch size indicates small latency while handling the test data online, and (2) multi, $N$-target domains, $\mathcal{D}_\mathcal{T} = \{\mathcal{D}_{T,i}\}_{i=1}^N$. Under this setting, each test batch $\mathcal{B}^T$ is drawn by one of the test domains in $\mathcal{D}_T$, where $\mathcal{D}_T$ may consist of a single target domain, multiple target domains, or mixture of target domains.

### 2.2 TEST-TIME NORMALIZATION LAYER

We denote an input of a BN layer as $\mathbf{z} \in \mathbb{R}^{BCHW}$, forming a mini-batch size of $B$. The mean and variance of $\mathbf{z}$ are $\mu$ and $\sigma^2$, respectively, which are computed as follows:

$$\mu_c = \frac{1}{BHW} \sum_b^B \sum_h^H \sum_w^W \mathbf{z}_{bchw}, \quad \sigma_c^2 = \frac{1}{BHW} \sum_b^B \sum_h^H \sum_w^W (\mathbf{z}_{bchw} - \mu_c)^2, \quad (1)$$

where $\mu$ and $\sigma^2$ are in $\mathbb{R}^C$, and $C$, $H$, and $W$ stand for the number of channels, dimension of height, and that of width, respectively. Based on $\mu$ and $\sigma^2$, the source statistics $\mu_s, \sigma_s^2 \in \mathbb{R}^C$ are usually estimated with exponential moving average over the training data.

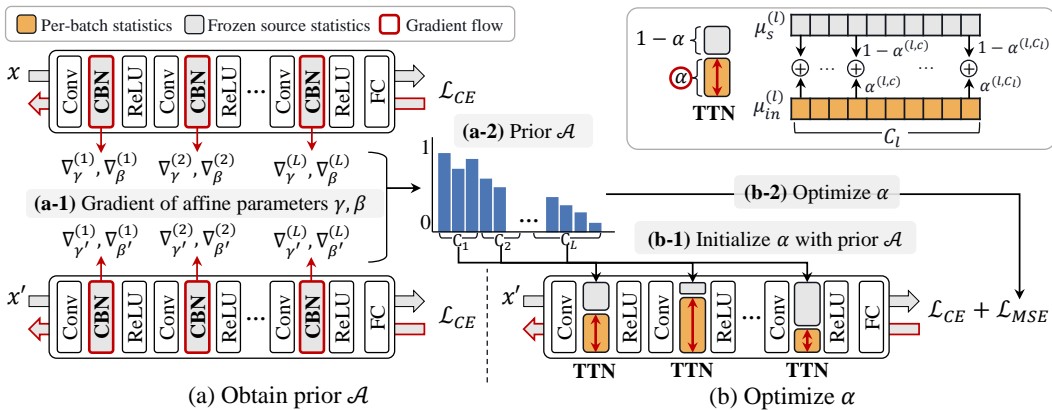

Figure 3: **Two stages in post-training phase.** **(a)** Given a pre-trained model, which uses CBN, and its training data, we obtain prior knowledge of each BN layer. **(a-1)** We first compute gradients of affine parameters in each BN layer from clean $x$ and augmented input $x'$ and obtain the gradient distance score (Eq. 4). **(a-2)** For BN layers with larger distance score, we put more importance on current batch statistics than source statistics (*i.e.,* large $\alpha$), and we define prior $\mathcal{A}$ accordingly (Eq. 5). **(b)** After obtaining prior $\mathcal{A}$, we substitute BN layers from CBN to TTN. **(b-1)** Initializing $\alpha$ with prior $\mathcal{A}$, **(b-2)** we optimize $\alpha$ using CE and MSE loss (Eq. 6) with augmented training data $x'$.

In BN layers, input $\mathbf{z}$ is first standardized with statistics $\mu$ and $\sigma^2$ and then is scaled and shifted with learnable parameters $\gamma$ and $\beta$ in $\mathbb{R}^C$. The standardization uses current input batch statistics during training and uses estimated source statistics $\mu_s$ and $\sigma_s^2$ at test time (Fig. 2(b)). To address domain shifts in test time, we adjust the source statistics by combining the source and the test mini-batch statistics (Singh & Shrivastava, 2019; Summers & Dinneen, 2019) with a learnable interpolating weight $\alpha \in \mathbb{R}^C$ ranges $[0, 1]$. Precisely, TTN standardizes a feature with

$$\tilde{\mu} = \alpha\mu + (1-\alpha)\mu_s, \quad \tilde{\sigma}^2 = \alpha\sigma^2 + (1-\alpha)\sigma_s^2 + \alpha(1-\alpha)(\mu - \mu_s)^2, \tag{2}$$

while using the same affine parameters, $\gamma$ and $\beta$. Note that we have different mixing ratios $\alpha_c$ for every layer and channel.

## 2.3 POST TRAINING

Like Choi et al. (2022), we introduce an additional training phase, the post-training (after pre-training but before testing), to optimize the mixing parameters $\alpha$ in Eq. 2 (Fig. 2(a)). Note that all parameters except $\alpha$ are frozen and we have access to the labeled source data during the post-training. We first obtain prior knowledge $\mathcal{A}$ of $\alpha$ by identifying which layers and their channels are sensitive to domain shifts. Then, we optimize $\alpha$ with the prior knowledge and an additional objective term. The overall procedure is depicted in Fig. 3 and the pseudocode is provided in appendix A.3.

**Obtain Prior $\mathcal{A}$.** To identify which BN layers and corresponding channels are sensitive to domain shifts, we simulate the domain shifts by augmenting[1] the clean image, *i.e.,* original training data, and make a pair of (clean $x$, domain-shifted $x'$) images, where the semantic information is shared. To analyze in which layer and channel the standardization statistics should be corrected, we consider the standardized features $\hat{z}^{(l,c)}$ of $z^{(l,c)}$, for a channel index $c$ at a layer $l$, whose input is clean $x$. We compare $\hat{z}^{(l,c)}$ to that of domain-shifted one, $\hat{z}'^{(l,c)}$ from $x'$. Since the pre-trained CBN uses the same $\mu_s^{(l,c)}$ and $\sigma_s^{(l,c)}$ for both inputs, the difference between $\hat{z}^{(l,c)}$ and $\hat{z}'^{(l,c)}$ is caused by the domain discrepancy between $x$ and $x'$. We argue that if the difference is significant, the parameter at $(l, c)$ is sensitive to the domain shift, *i.e.,* intensely affected by the domain shift, and hence the standardization statistics at $(l, c)$ should be adapted towards the shifted input.

Drawing inspiration from Choi et al. (2022), we measure the domain-shift sensitivity by comparing gradients. Since the standardized feature $\hat{z}$ is scaled and shifted by $\gamma$ and $\beta$ in each BN layer, we compare the gradients of affine parameters $\gamma$ and $\beta$, $\nabla_\gamma$ and $\nabla_\beta$, respectively, to measure the dissimilarity of $\hat{z}$ and $\hat{z}'$. As described in Fig. 3(a-1), we collect the $\nabla_\gamma$ and $\nabla_\beta$ using cross-entropy

---

[1]It is noteworthy that the post-training phase is robust to the choice of data augmentation types. Ablation study results and discussions are provided in the appendix B.4.

loss, $\mathcal{L}_{\text{CE}}$. To this end, we introduce a *gradient distance score*, $d^{(l,c)} \in \mathbb{R}$ for each channel $c$ at layer $l$ as follows:

$$s = \frac{1}{N}\sum_{i=1}^{N}\frac{g_i \cdot g_i'}{\|g_i\|\|g_i'\|}, \tag{3}$$

$$d^{(l,c)} = 1 - \frac{1}{2}\big(s_\gamma^{(l,c)} + s_\beta^{(l,c)}\big), \tag{4}$$

where $(g, g')$ is $(\nabla_\gamma^{(l,c)}, \nabla_{\gamma'}^{(l,c)})$ and $(\nabla_\beta^{(l,c)}, \nabla_{\beta'}^{(l,c)})$ for $s_\gamma^{(l,c)}$ and $s_\beta^{(l,c)}$, respectively, $N$ is the number of training data, and the resulting $d^{(l,c)} \in [0, 1]$. Once we obtain $s_\gamma$ and $s_\beta$ from Eq. 3, we conduct min-max normalization over all $s_\gamma^{(l,c)}$ and $s_\beta^{(l,c)}$, before computing Eq. 4.

To magnify the relative difference, we take the square as a final step and denote the result as a prior $\mathcal{A}$ (Fig. 3(a-2)):

$$\mathcal{A} = [d^{(1,\cdot)}, d^{(2,\cdot)}, \ldots, d^{(L,\cdot)}]^2, \tag{5}$$

where $d^{(l,\cdot)} = [d^{(l,c)}]_{c=1}^{C_l}$.

**Optimize $\alpha$.** The goal of optimizing $\alpha$ is to make the combined statistics correctly standardize the features when the input is sampled from an arbitrary target domain. After obtaining the prior $\mathcal{A}$, we replace CBN with TTN layers while keeping the affine parameters. Then, we initialize the interpolating weights $\alpha$ with $\mathcal{A}$, which represents in which layer and channel the standardization statistics need to be adapted using test batch statistics (see Fig. 3(b-1)). To simulate distribution shifts, we use augmented training data. Expecting the model to make consistent predictions either given clean or augmented inputs, we use cross-entropy loss $\mathcal{L}_{\text{CE}}$. Furthermore, to prevent $\alpha$ from moving too far from the initial value $\mathcal{A}$, we use mean-squared error loss $\mathcal{L}_{\text{MSE}}$ between $\alpha$ and the prior $\mathcal{A}$, *i.e.,* $\mathcal{L}_{\text{MSE}} = \|\alpha - \mathcal{A}\|^2$ as a constraint. Total loss $\mathcal{L}$ can be written as $\mathcal{L} = \mathcal{L}_{\text{CE}} + \lambda \mathcal{L}_{\text{MSE}}$ (6), where $\lambda$ is a weighting hyperparameter (Details are provided in the appendix A.1 & B.1).

## 3 EXPERIMENTS

In image classification, we evaluate TTN for corruption robustness in realistic evaluation settings, *i.e.,* where the test batch size can be variant and where the target domain can be either stationary, continuously changing, or mixed with multiple domains. Additionally, we further validate TTN on domain generalization benchmarks incorporating natural domain shifts (*e.g.,* changes in camera sensors, weather, time, and region) in semantic segmentation.

### 3.1 EXPERIMENTAL SETUP

Given models pre-trained on clean source data, we optimize TTN parameter $\alpha$ with the augmented source data in the post-training phase. Afterward, we evaluate our post-trained model on the corrupted target data. **Implementation details are provided in the appendix A.1.**

**Datasets and models.** We use corruption benchmark datasets CIFAR-10/100-C and ImageNet-C, which consist of 15 types of common corruptions at five severity levels (Hendrycks & Dietterich, 2018). Each corruption is applied to test images of the clean datasets (CIFAR-10/100 and ImageNet). We use a training set of the clean dataset for post-training and the corrupted dataset for evaluation. As backbone models, we used WideResNet-40-2 (Hendrycks et al., 2019) trained on CIFAR-10/100, and ResNet-50 (He et al., 2016) trained on ImageNet. To validate our method in semantic segmentation, we conduct experiments on Cityscapes (Cordts et al., 2016), BDD-100K (Yu et al., 2020), Mapillary (Neuhold et al., 2017), GTAV (Richter et al., 2016), and SYNTHIA (Ros et al., 2016) datasets, in accordance with the experimental setup for domain generalization proposed in RobustNet (Choi et al., 2021).

**Baselines.** To demonstrate that TTN successfully controls the trade-off between CBN and TBN, we compare TTN with (1) AdaptiveBN (Schneider et al., 2020), (2) $\alpha$-BN (You et al., 2021) and (3) MixNorm (Hu et al., 2021), which combines or takes the moving average of the source and the test batch statistics with a pre-defined hyperparameter (*i.e.,* a constant $\alpha$). The following baselines are suggested on top of TBN (Nado et al., 2020); (4) TENT (Wang et al., 2020) optimizes BN affine parameters via entropy minimization. (5) SWR (Choi et al., 2022) updates the entire model parameters considering the domain-shift sensitivity. (6) CoTTA (Wang et al., 2022) ensembles the output of

Table 1: **Single domain adaptation on corruption benchmark.** Error rate ($\downarrow$) averaged over 15 corruptions with severity level 5 using WideResNet-40-2 as a backbone for each test batch size. We used reported results of MixNorm with fixed parameters from the original paper and denoted as $*$. In appendix B.3, we provide variants of TTN, which show stronger performance for small test batch.

| | Method | CIFAR-10-C | | | | | | | CIFAR-100-C | | | | | | |
|---|---|---|---|---|---|---|---|---|---|---|---|---|---|---|---|
| | | 200 | 64 | 16 | 4 | 2 | 1 | Avg. | 200 | 64 | 16 | 4 | 2 | 1 | Avg. |
| | Source (CBN) | 18.27 | 18.27 | 18.27 | 18.27 | 18.27 | 18.27 | 18.27 | 46.75 | 46.75 | 46.75 | 46.75 | 46.75 | 46.75 | 46.75 |
| Norm | TBN | 14.49 | 15.02 | 17.10 | 26.28 | 35.65 | 90.00 | 33.09 | 39.25 | 40.21 | 44.03 | 59.10 | 80.65 | 99.04 | 60.38 |
| | AdaptiveBN | 12.21 | 12.31 | 12.89 | 14.51 | 15.79 | 16.14 | 13.98 | 36.56 | 36.85 | 38.19 | 41.18 | 43.26 | **44.01** | **40.01** |
| | $\alpha$-BN | 13.78 | 13.77 | 13.89 | 14.54 | **15.16** | 15.47 | 14.44 | 39.72 | 39.85 | 39.99 | 41.34 | **42.66** | 45.64 | 41.53 |
| | MixNorm* | 13.85 | 14.41 | 14.23 | 14.60 ($B$=5) | - | 15.09 | 14.44 | - | - | - | - | - | - | - |
| | **Ours (TTN)** | **11.67** | **11.80** | **12.13** | **13.93** | 15.83 | 17.99 | **13.89** | **35.58** | **35.84** | **36.73** | **41.08** | 46.67 | 57.71 | 42.27 |
| Optim. | TENT | 12.08 | 14.78 | 16.90 | 25.61 | 35.69 | 90.00 | 32.51 | 35.52 | 39.90 | 43.78 | 59.02 | 80.68 | 99.02 | 59.65 |
| | **+Ours (TTN)** | **11.28** | **11.52** | **12.04** | **13.95** | **15.84** | **17.94** | **13.77** | **35.16** | **35.57** | **36.55** | **41.18** | **46.63** | **58.33** | **42.24** |
| | SWR | 10.26 | 13.51 | 16.61 | 27.33 | 40.48 | 90.04 | 33.04 | **32.68** | 37.41 | 43.15 | 59.90 | 87.07 | 99.05 | 59.88 |
| | **+Ours (TTN)** | **9.92** | **11.77** | **13.41** | **18.02** | **24.09** | **61.56** | **23.13** | 32.86 | **35.13** | **38.66** | **49.80** | **60.72** | **80.90** | **49.68** |

Table 2: **Continuously changing domain adaptation on corruption benchmark.** Error rate ($\downarrow$) averaged over 15 corruptions with severity level 5 using WideResNet-40-2 as backbone for each test batch size. We omitted 'Norm' methods results in this table since they are eqaul to that of Table 1.

| | Method | CIFAR-10-C | | | | | | | CIFAR-100-C | | | | | | |
|---|---|---|---|---|---|---|---|---|---|---|---|---|---|---|---|
| | | 200 | 64 | 16 | 4 | 2 | 1 | Avg. | 200 | 64 | 16 | 4 | 2 | 1 | Avg. |
| | Source (CBN) | 18.27 | 18.27 | 18.27 | 18.27 | 18.27 | 18.27 | 18.27 | 46.75 | 46.75 | 46.75 | 46.75 | 46.75 | 46.75 | 46.75 |
| | **Ours (TTN)** | **11.67** | **11.80** | **12.13** | **13.93** | **15.83** | **17.99** | **13.89** | **35.58** | **35.84** | **36.73** | **41.08** | **46.67** | **57.71** | **42.27** |
| Optim. | CoTTA | 12.46 | 14.60 | 21.26 | 45.69 | 58.87 | 90.00 | 40.48 | 39.75 | 42.20 | 52.94 | 73.69 | 87.66 | 98.99 | 65.87 |
| | TENT | 12.54 | 13.52 | 15.69 | 26.23 | 35.77 | 90.00 | 32.29 | **36.11** | 37.90 | 43.78 | 58.71 | 81.76 | 99.04 | 59.55 |
| | **+Ours (TTN)** | **11.44** | **11.60** | **12.08** | **16.14** | **18.36** | **22.40** | **15.33** | 43.50 | **37.60** | **38.28** | **44.60** | **54.29** | **80.63** | **49.82** |
| | SWR | 11.04 | 11.53 | 13.90 | 23.99 | 34.02 | 90.00 | 30.75 | 34.16 | 35.79 | 40.71 | 58.15 | 80.55 | 99.03 | 62.56 |
| | **+Ours (TTN)** | **10.09** | **10.51** | **11.28** | **14.29** | **16.67** | **84.12** | **24.49** | **33.09** | **34.07** | **36.15** | **42.41** | **53.63** | **93.08** | **48.74** |

augmented test inputs, updates the entire model parameters using a consistency loss between student and teacher models, and stochastically restores the pre-trained model. We refer to TBN, (1), (2), and (3) as *normalization*-based methods (Norm), the other as *optimization*-based methods (Optim.), and denote the pre-trained model using CBN as 'source'.

**Evaluation scenarios.** To show that TTN performs robust on various test batch sizes, we conduct experiments with test batch sizes of 200, 64, 16, 4, 2, and 1. We evaluate our method in three evaluation scenarios; *single*, *continuously changing*, and *mixed* domain adaptation. In the single domain adaptation, the model is optimized for one corruption type and then reset before adapting to the subsequent corruption, following the evaluation setting from TENT and SWR. In the continuously changing adaptation (Wang et al., 2022), the model is continuously adapted to 15 corruption types (w/o the reset), which is more realistic because it is impractical to precisely indicate when the data distribution has shifted in the real world. Finally, to simulate the non-stationary target domain where various domains coexist, we evaluate methods in the mixed domain adaptation setting, where a single batch contains multiple domains. We use a severity level of 5 (Hendrycks & Dietterich, 2018) for all experiments. It is noteworthy that we use a single checkpoint of TTN parameter $\alpha$ for each dataset across all experimental settings.

## 3.2 EXPERIMENTS ON IMAGE CLASSIFICATION

Tables 1, 2, and 3 show error rates on corruption benchmark datasets in three different evaluation scenarios; single domain, continuously changing, and mixed domain adaptation, respectively. Note that the performance of normalization-based methods in the single (Table 1) and in the continuously changing (Table 2) settings are identical. Tables 4 and 5 show the adaptation performance on the source and class imbalanced target domains, respectively. **More results and discussions are provided in the appendix B, importantly, including results on ImageNet-C (B.5).**

**Robustness to practical settings.** In Table 1, 2, and 3, TTN and TTN applied methods show robust performance over the test batch size ranges from 200 to 1. Comparing with normalization-based baselines, we demonstrate that TTN, which uses channel-wisely optimized combining rate $\alpha$, shows better results than defining $\alpha$ as a constant hyperparameter, which can be considered as a special

Table 3: **Mixed domain adaptation on corruption benchmark.** Error rate ($\downarrow$) of mixed domain with severity level 5 using WideResNet-40-2 as backbone for each test batch size. We used the reported results of MixNorm with fixed parameters from the original paper and denoted them as $*$.

| | Method | CIFAR-10-C | | | | | | | CIFAR-100-C | | | | | | |
|---|---|---|---|---|---|---|---|---|---|---|---|---|---|---|---|
| | | 200 | 64 | 16 | 4 | 2 | 1 | Avg. | 200 | 64 | 16 | 4 | 2 | 1 | Avg. |
| | Source (CBN) | 18.27 | 18.27 | 18.27 | 18.27 | 18.27 | 18.27 | 18.27 | 46.75 | 46.75 | 46.75 | 46.75 | 46.75 | 46.75 | 46.75 |
| Norm | TBN | 14.99 | 15.29 | 17.38 | 26.65 | 35.59 | 90.00 | 33.31 | 39.88 | 40.48 | 43.73 | 59.11 | 80.30 | 98.91 | 60.40 |
| | AdaptiveBN | 12.62 | 12.48 | 12.97 | 14.59 | 15.74 | 16.02 | 14.07 | 36.88 | 36.86 | 38.49 | 41.43 | 43.38 | 44.31 | 40.23 |
| | $\alpha$-BN | 13.78 | 13.78 | 13.99 | 14.61 | **15.07** | **15.20** | 14.41 | 40.25 | 40.11 | 40.47 | 41.64 | **42.39** | **43.81** | **41.45** |
| | MixNorm* | 18.80 | 18.80 | 18.80 | 18.80 | 18.80 | 18.80 | 18.80 | - | - | - | - | - | - | - |
| | **Ours (TTN)** | **12.16** | **12.19** | **12.34** | **13.96** | 15.55 | 17.83 | **14.00** | **36.24** | **36.23** | **36.85** | **41.01** | 45.85 | 55.52 | 41.95 |
| Optim. | TENT | 14.33 | 14.97 | 17.30 | 26.07 | 35.37 | 90.00 | 33.01 | 39.36 | 40.01 | 43.33 | 58.98 | 80.55 | 98.92 | 60.19 |
| | **+Ours (TTN)** | **12.02** | **12.04** | **12.20** | **13.77** | **15.42** | **16.40** | **13.64** | **36.29** | **36.23** | **36.89** | **41.38** | **46.65** | **57.95** | **42.57** |
| | SWR | 13.24 | 13.06 | 16.57 | 26.08 | 38.65 | 91.03 | 59.54 | 37.84 | 37.93 | 44.37 | 59.50 | 78.66 | 98.95 | 33.10 |
| | **+Ours (TTN)** | **11.89** | **11.65** | **13.37** | **17.05** | **23.50** | **64.10** | **50.29** | **36.49** | **36.51** | **39.60** | **46.20** | **58.20** | **84.76** | **23.59** |

case of TTN; TBN and $\alpha$-BN corresponds to $\alpha = 1$ and $0.1$, respectively. More comparisons with different constant $\alpha$ are provided in the appendix B.2. It is noteworthy that TTN as a stand-alone method favorably compares with optimization-based baselines in all three scenarios.

Table 4: **Source domain adaptation.** Error rate ($\downarrow$) on CIFAR-10 using WideResNet-40-2.

| | Method | Test batch size | | | | | | Avg. |
|---|---|---|---|---|---|---|---|---|
| | | 200 | 64 | 16 | 4 | 2 | 1 | |
| | Source (CBN) | 4.92 | 4.92 | 4.92 | 4.92 | 4.92 | 4.92 | 4.92 |
| Norm | TBN | 6.41 | 6.60 | 8.64 | 17.65 | 26.08 | 90.00 | 25.90 |
| | **Ours (TTN)** | **4.88** | **5.11** | **5.35** | **7.27** | **9.45** | **9.96** | **7.00** |
| Optim. | TENT | 6.15 | 6.45 | 8.61 | 17.61 | 26.20 | 90.00 | 32.2 |
| | **+Ours (TTN)** | **4.93** | **5.11** | **5.32** | **7.22** | **9.38** | **10.21** | **7.02** |
| | SWR | 5.63 | 6.01 | 8.25 | 17.49 | 26.32 | 90.00 | 25.62 |
| | **+Ours (TTN)** | **4.79** | **5.02** | **5.51** | **6.68** | **7.91** | **9.34** | **6.54** |

Table 5: **Class imbalanced target domain.** Error rate ($\downarrow$) averaged over 15 corruptions of CIFAR-10-C with severity level 5 using WideResNet-40-2. Details are provided in the appendix A.2.

| Method | Test batch size | | | | | | Avg. |
|---|---|---|---|---|---|---|---|
| | 200 | 64 | 16 | 4 | 2 | 1 | |
| Source (CBN) | 18.27 | 18.27 | 18.27 | 18.27 | 18.27 | 18.27 | 18.27 |
| TBN | 77.60 | 76.66 | 77.72 | 78.59 | 77.84 | 90.00 | 79.74 |
| **Ours (TTN)** | **35.75** | **35.13** | **34.92** | **32.51** | **28.60** | **17.99** | **30.82** |

**Adopting TTN improves other TTA methods.** We compare optimization-based methods with and without TTN layers. Since TENT, SWR, and CoTTA optimize model parameters on top of using TBN layers, they also suffer from performance drops when the test batch size becomes small. Adopting TTN reduces the dependency on large test batch size, *i.e.,* makes robust to small batch size, and even improves their performance when using large test batch. Furthermore, in continual (Table 2) and mixed domain (Table 3) adaptation scenario, TENT and SWR shows higher error rate than in single domain (Table 1) adaptation. We interpret that because they update the model parameters based on the current output and predict the next input batch using the updated model, the model will not perform well if the consecutive batches have different corruption types (*i.e.,* mixed domain adaptation). Moreover, the error from the previous input batch propagates to the future input stream, and thus they may fall apart rapidly once they have a strongly wrong signal, which can happen in continual adaptation (*i.e.,* long-term adaptation without resetting). Applying TTN significantly accelerates their model performance regardless of the evaluation scenarios.

**TTN preserves knowledge on source domain.** In practice, data driven from the source domain (or a merely different domain) can be re-encountered in test time. We used clean domain test data in the single domain adaptation scenario to show how TTN and other TTA methods adapt to the seen source domain data (but unseen instance). As shown in Table 4, all baseline methods using TBN layers, show performance drops even with large batch sizes. We can conclude that it is still better to rely on source statistics collected from the large training data than using only current input statistics, even if its batch size is large enough to obtain reliable statistics (*i.e.,* 200). However, since TTN utilizes source statistics while leveraging the current input, TTN itself and TTN adopted methods well preserve the source knowledge compared to the TBN-based methods. With a batch size of 200, we observe that combining the source and a current test batch statistics outperforms the source model (see 3rd row of Table 4).

**TTN is robust to class imbalanced scenario.** Heavily depending on current test batch statistics are especially vulnerable when the class labels are imbalanced (Boudiaf et al., 2022; Gong et al., 2022). To simulate this situation, we sorted test images in class label order and then sampled test batches following the sorted data order. In Table 5, we observe that TTN is more robust to the class imbalanced scenario than utilizing only test batch statistics (*i.e.,* TBN). As explained in Section 3.5,

Table 6: **Adaptation on DG benchmarks in semantic segmentation.** mIoU($\uparrow$) on four unseen domains with test batch size of 2 using ResNet-50 based DeepLabV3+ as a backbone.

| | Method (Cityscapes→) | BDD-100K | Mapillary | GTAV | SYNTHIA | Cityscapes |
|---|---|---|---|---|---|---|
| | Source (Chen et al., 2018) | 43.50 | 54.37 | 43.71 | 22.78 | 76.15 |
| Norm | TBN | 43.12 | 47.61 | 42.51 | 25.71 | 72.94 |
| | **Ours (TTN)** | **47.40** | **56.92** | **44.71** | **26.68** | **75.09** |
| Optim. | TENT | 43.30 | 47.80 | 43.57 | 25.92 | 72.93 |
| | **+ Ours (TTN)** | **47.89** | **57.84** | **46.18** | **27.29** | **75.04** |
| | SWR | 43.40 | 47.95 | 42.88 | 25.97 | 72.93 |
| | **+ Ours (TTN)** | **48.85** | **59.09** | **46.71** | **29.16** | **74.89** |

we are putting more importance on CBN than TBN, where semantic information is mainly handled, *i.e.,* in deeper layers, so we can understand that TTN is less impacted by skewed label distribution.

## 3.3 EXPERIMENTS ON SEMANTIC SEGMENTATION

We additionally conduct experiments on domain generalization (DG) benchmarks (Choi et al., 2021; Pan et al., 2018) for semantic segmentation, including natural domain shifts (*e.g.,* Cityscapes→BDD-100K), to demonstrate the broad applicability of TTN. Table 6 shows the results of evaluating the ResNet-50-based DeepLabV3+ (Chen et al., 2018) model trained on the Cityscapes training set using the validation set of real-world datasets such as Cityscapes, BDD-100K, and Mapillary, and synthetic datasets including GTAV and SYNTHIA. We employ a test batch size of 2 for test-time adaptation in semantic segmentation. We observe that even when exploiting test batch statistics for standardization in BN layers (TBN) or updating the model parameters on top of TBN (TENT, SWR) does not improve the model performance (*i.e.,* perform worse than the source model), adopting TTN helps the model make good use of the strength of the test batch statistics. Implementation details and additional results are provided in the appendix A.1 and B.7, respectively.

## 3.4 ABLATION STUDIES

**Prior $\mathcal{A}$ regularizes $\alpha$ to be robust to overall test batch sizes.** We conduct an ablation study on the importance of each proposed component, *i.e.,* initializing $\alpha$ with prior $\mathcal{A}$, optimizing $\alpha$ using CE and MSE losses, and the results are shown in Table 7. Using $\mathcal{A}$ for initialization and MSE loss aims to optimize $\alpha$ following our intuition that we discussed in Section 2.3. Optimizing $\alpha$ using CE loss improves the overall performance, but without regularizing with MSE loss, $\alpha$ may overfit to large batch size (rows 2 & 3). Initialization with $\mathcal{A}$ or not does not show a significant difference, but $\mathcal{A}$ provides a better starting point than random initialization when comparing the left and right of the 2nd row. We observe that when using MSE loss, regardless of initialization using $\mathcal{A}$, the optimized $\alpha$ sufficiently reflects our intuition resulting in a low error rate to overall batch sizes (row 3).

Table 7: **Ablation study on importance of each component**

| Method | | | Test batch size | | | | | | Avg. | Method | | | Test batch size | | | | | | Avg. |
|---|---|---|---|---|---|---|---|---|---|---|---|---|---|---|---|---|---|---|---|
| Init. | CE | MSE | 200 | 64 | 16 | 4 | 2 | 1 | | Init. | CE | MSE | 200 | 64 | 16 | 4 | 2 | 1 | |
| - | - | ✓ | 13.36 | 13.43 | 13.85 | 15.50 | 17.43 | 20.07 | 15.61 | ✓ | - | - | 13.37 | 13.43 | 13.85 | 15.50 | 17.44 | 20.07 | 15.61 |
| - | ✓ | - | **11.64** | **11.73** | 12.26 | 14.46 | 16.94 | 19.88 | 14.49 | ✓ | ✓ | - | 11.73 | 11.82 | 12.23 | 14.18 | 16.41 | 19.27 | 14.27 |
| - | ✓ | ✓ | **11.64** | 11.78 | **12.21** | **13.97** | 15.86 | 18.00 | 13.91 | ✓ | ✓ | ✓ | **11.67** | **11.80** | 12.13 | 13.93 | 15.83 | 17.99 | 13.89 |

## 3.5 VISUALIZATION OF $\alpha$

Fig. 4 shows the visualization of optimized $\alpha$ for CIFAR-10 using WideResNet-40-2. We observe that $\alpha$ decreases from shallow to deep layers (left to right), which means CBN is more active in deeper layers, and TBN is vice versa. As shown in Table 4 and 6, CBN employing source statistics is superior to TBN when the distribution shift between source and target domains is small. Assuming that the $\alpha$ we obtained is optimal, we can conjecture that CBN is more active (*i.e.,* $\alpha$ closer to 0) in deeper layers because domain information causing the distribution shift has been diminished. In contrast, TBN has a larger weight (*i.e.,* $\alpha$ closer to 1) in shallower layers since the domain information induces a large distribution shift. This interpretation is consistent with the observations of previous studies (Pan et al., 2018; Wang et al., 2021; Kim et al., 2022a) that style information mainly exists in shallower layers, whereas only content information remains in deeper layers.

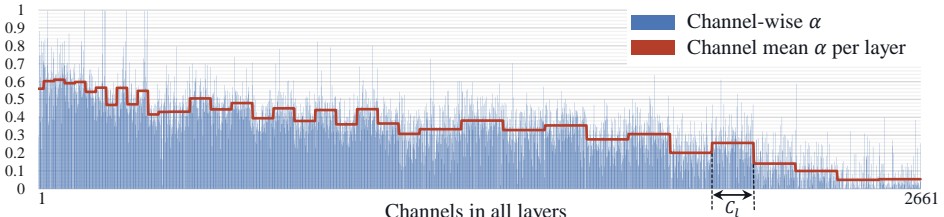

Figure 4: **Optimized** $\alpha$. x- and y-axes indicate all channels in order from shallow to deep layers and the interpolating weight $\alpha$, respectively. $C_l$ denotes the channel size of layer $l$.

## 4    RELATED WORK

Test-time adaptation/training (TTA) aims to adapt models towards test data to overcome the performance degradation caused by distribution shifts (Sun et al., 2020; Wang et al., 2020). There are other related problems, unsupervised domain adaptation (UDA) (Sun & Saenko, 2016; Ganin et al., 2016) and source-free domain adaptation (SFDA) (Liang et al., 2020; Huang et al., 2021; Liu et al., 2021). Both UDA and SFDA have access to sufficiently large enough unlabeled target datasets, and their objective is to achieve high performance on that particular target domain. Unlike UDA and SFDA, TTA utilizes test data in an online manner. There are two key factors of recent approaches: adapting standardization statistics in normalization layers and adapting model parameters.

**Normalization in Test Time.** Nado et al. (2020) suggested prediction-time BN, which uses test batch statistics for standardization and Schneider et al. (2020) introduced to adapt BN statistics by combining source and test batch statistics considering the the test batch size to mitigate the intermediate covariate shift. In this paper, we refer to the former method as TBN. Similarly, You et al. (2021) and Khurana et al. (2021) mixed the statistics using predefined hyperparameter. Also, Mirza et al. (2022) and Hu et al. (2021) adapted the statistics using moving average while augmenting the input to form a pseudo test batch when only a single instance is given. The primary difference with the existing approaches is that we consider the channel-wise domain-shift sensitivity of BN layers to optimize the interpolating weights between CBN and TBN. Concurrently, Zou et al. (2022) proposed to adjust the standardization statistics using a learnable calibration strength and showed its effectiveness focusing on the semantic segmentation task.

**Optimization in Test Time.** TENT (Wang et al., 2020), SWR (Choi et al., 2022), and CoTTA (Wang et al., 2022) updated model parameters while using TBN. TENT optimized affine parameters in BN layers using entropy minimization while freezing the others. To maximize the adaptability, SWR updated the entire model parameters minimizing the entropy loss based on the domain-shift sensitivity. To stabilize the adaptation in continuously changing domains, CoTTA used consistency loss between student and teacher models and stochastically restored random parts of the pre-trained model. Liu et al. (2021) and Chen et al. (2022) suggested to update the model through contrastive learning.

We focus on correcting the standardization statistics using domain-shift aware interpolating weight $\alpha$. Similar to Choi et al. (2022), we measure the domain-shift sensitivity by comparing gradients. The principle difference is that we use channel-wise sensitivity when optimizing $\alpha$ in post-training, while SWR used layer-wise sensitivity regularizing the entire model update in test time.

## 5    CONCLUSIONS

This paper proposes TTN, a novel domain-shift aware batch normalization layer, which combines the benefits of CBN and TBN that have a trade-off relationship. We present a strategy for mixing CBN and TBN based on the interpolating weight derived from the optimization procedure utilizing the sensitivity to domain shift and show that our method significantly outperforms other normalization techniques in various realistic evaluation settings. Additionally, our method is highly practical because it can complement other optimization-based TTA methods. The oracle mixing ratio between CBN and TBN can vary depending on the domain gap difference. However, our proposed method employs a fixed mixing ratio during test time, where the mixing ratio is optimized before model deployment. If we could find the optimal mixing ratio according to the distribution shift during test time, we can expect further performance improvement. We consider it as future work. In this regard, our efforts encourage this field to become more practical and inspire new lines of research.

## REPRODUCIBILITY STATEMENT

To ensure the reproducibility of our method, we provide the experimental setup in Section 3.1. Moreover, the details on implementation and evaluation settings can be found in the appendix A.1 and A.2, respectively. The pseudocode for overall training and testing scheme is provided in the appendix A.3. Together with related references and publicly available codes, we believe our paper contains sufficient information for reimplementation.

## ACKNOWLEDGEMENT

We would like to thank Kyuwoong Hwang, Simyung Chang, and Seunghan Yang of the Qualcomm AI Research team for their valuable discussions.

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

# A   APPENDIX: FOR REPRODUCIBILITY

This section provides supplemental material for Section 2 and 3.1.

## A.1   IMPLEMENTATION DETAILS

**Datasets and Models.** For CIFAR-10/100-C, we optimized $\alpha$ using augmented CIFAR-10/100 training set on the pre-trained WideResNet-40-2 (WRN-40) (Hendrycks et al., 2019). For ImageNet-C, we used augmented ImageNet training set (randomly sampled 64000 instances per epoch) on the pre-trained ResNet-50.

**Augmentation.** Following the setting of SWR (Choi et al., 2022), we used color jittering, random invert and random grayscale when obtaining the prior $\mathcal{A}$. When optimizing $\alpha$, we followed the augmentation choice of CoTTA (Wang et al., 2022), which are color jittering, padding, random affine, gaussian blur, center crop and random horizontal flip. We excluded for gaussian noise to avoid any overlap with corruption type of common corruptions (Hendrycks & Dietterich, 2018). For ImageNet, we only used the same augmentation both for obtaining $\mathcal{A}$ and optimizing $\alpha$ following the SWR augmentation choices.

**Post-training.** When obtaining prior, we used randomly selected 1024 samples from the training set, following the setting of SWR. We used Adam (Kingma & Ba, 2015) optimizer using a learning rate (LR) of 1e-3, which is decayed with cosine schedule (Loshchilov & Hutter, 2017) for 30 epochs and used 200 training batch for CIFAR-10/100. For ImageNet, we lowered LR to 2.5e-4 and used 64 batch size. For semantic segmentation task, we trained TTN using Cityscapes training set, and training batch size of 2. We resized the image height to 800 while preserving the original aspect ratio Cordts et al. (2016). We can terminate the training when MSE loss saturates (We observed that $\alpha$ does not show significant difference after the MSE loss is saturated). We used the weighting hyperparmeter to MSE loss $\lambda$ as 1.

**Test time.** For AdaptiveBN, which adjusts the interpolating weight using two factors: hyperparameter $N$ and test batch size $n$, we followed the suggested $N$, which is empirically obtained best hyperparameter from the original paper, for each $n$ (Figure 11, ResNet architecture from Schneider et al. (2020)). In detail, we set $N$ as 256, 128, 64, 32, and 16 for test batch size 200, 64, 16, 4, 2, and 1, which yields $\alpha$ as 0.44, 0.33, 0.2, 0.11, 0.06, and 0.06, respectively.

For optimization-based TTA methods, we followed default setting in TENT, SWR, and CoTTA for test-time adaptation. We used LR of 1e-3 to test batch size of 200 for CIFAR-10/100-C in single domain (TENT, SWR) and continuously changing domain (CoTTA) scenarios. To avoid rapid error accumulation, we lowered LR to 1e-4 for TENT and SWR in continual and mixed domain scenarios. Moreover, we updated model parameters after accumulating the gradients for 200 samples for CIFAR-10/100-C. In other words, we compute gradients per batch, but update, *i.e.,* `optimizer.step()`, after seeing 200 data samples. Exceptionally, we used LR of 5e-5 for SWR and SWR+TTN in mixed domain setting. Additionally, in continuously changing and mixed domain scenarios, we used the stable version of SWR, which updates the model parameter based on the frozen source parameters instead of previously updated parameters (original SWR).

For semantic segmentation, we set the test batch size as 2 and learning rate for optimization-based methods as 1e-6 for all datasets. For SWR, we set the importance of the regularization term $\lambda_r$ as 500. The other hyperparameters are kept the same as Choi et al. (2022) choices. For all test-time adaptation, we used constant learning rate without scheduling.

## A.2   EVALUATION SCENARIO DETAILS

**Class imbalanced setting.** In the main paper Table 5, we show the results under class imbalanced settings. In the setting, we sorted the test dataset of each corruption in the order of class labels, i.e., from class 0 to 9 for CIFAR-10-C. Then, we comprised test batches following the sorted order. Therefore, most batches consist of single class input data, which leads to biased test batch statistics. For larger batch size, the statistics are more likely to be extremely skewed, and that explains why error rates are higher with larger batch sizes than with the small ones.

A.3   PSEUDOCODE

Pseudocode for post-training, *i.e.,* obtaining $\mathcal{A}$ and optimizing $\alpha$, is provided in Algorithms 1 and 2, respectively, and that for test time is in Algorithm 3. Moreover, we provide PyTorch-friendly pseudocode for obtaining $\mathcal{A}$ in Listing 1. Please see Section 2 for equations and terminologies used in the algorithms.

---

**Algorithm 1** Obtain prior $\mathcal{A}$

---

1: **Require**: Pre-trained model $f_\theta$; source training data $\mathcal{D}_S = (X, Y)$
2: **Output**: Prior $\mathcal{A}$
3: **for** all $(x, y)$ in $\mathcal{D}_S$ **do**
4:     Augment $x$: $x'$
5:     Collect gradients $(\nabla_\gamma, \nabla_{\gamma'})$ and $(\nabla_\beta, \nabla_{\beta'})$ from $f_\theta$ using clean $x$ and augmented $x'$
6: **end for**
7: Compute gradient distance score $d$ using Eq. 4
8: Define prior $\mathcal{A}$ using Eq. 5

---

**Algorithm 2** Post-train

---

1: **Require**: Pre-trained model $f_\theta$; source training data $\mathcal{D}_S = (X, Y)$;
    step size hyperparameter $\eta$; regularization weight hyperparameter $\lambda$
2: **Output**: Optimized interpolating weight $\alpha$
3: Obtain prior $\mathcal{A}$ using Algorithm 1
4: Initialize $\alpha$ with prior $\mathcal{A}$
5: Replace all BN layers of $f_\theta$ to TTN layers using $\alpha$ and Eq. 2
6: **while** not done **do**
7:     Sample minibatches $\mathcal{B}^S$ from $\mathcal{D}_S$
8:     **for** all minibatches **do**
9:         Augment all $x$ in $\mathcal{B}^S$: $x'$
10:         Evaluate $\nabla_\alpha \mathcal{L}$ using $f_\theta$ given $\{(x_i', y_i)\}_{i=1}^{|\mathcal{B}^S|}$ using Eq. 6 while adapting standardization statistics using Eq. 2
11:         Update $\alpha \leftarrow \alpha - \eta \nabla_\alpha \mathcal{L}$
12:     **end for**
13: **end while**

---

**Algorithm 3** Inference (Test time)

---

1: **Require**: Pre-trained model $f_\theta$; optimized $\alpha$, target test data $\mathcal{D}_T = (X)$;
2: Replace all BN layers of $f_\theta$ to TTN layers using $\alpha$ and Eq. 2
3: Sample minibatches $\mathcal{B}^T$ from $\mathcal{D}_T$
4: **for** all minibatches **do**
5:     Make prediction using $f_\theta$ given $\mathcal{B}^T$ while adapting standardization statistics using Eq. 2
6: **end for**

---

```python
def obtain_prior(model, train_data):
  # make weight and bias of BN layers requires_grad=True, otherwise False
  params = {n: p for n, p in model.named_parameters() if p.requires_grad}

  grad_sim = {}
  for x, y in train_data:
    # collect gradients for clean and augmented input
    grad_org = collect_grad(model, x, y, params)
    grad_aug = collect_grad(model, augment(x), y, params)

    # compute grad similarity
    for n, p in params.items():
      grad_sim[n].data = cosine_sim(grad_org, grad_aug)
```

```
15    # average over data samples
16    for n, p in params.items():
17      grad_sim[n].data /= len(train_data)
18
19    # min max normalization
20    max_grad = get_max_value(grad_sim) # scalar
21    min_grad = get_min_value(grad_sim) # scalar
22    grad_dist = {}
23    for n, p in grad_sim.items():
24      grad_dist[n] = (p - min_grad) / (max_grad - min_grad)
25
26    prior = []
27    j = 0
28    # integrate gradients of weight(gamma) and bias(beta) for each BN layer
29    for n, p in grad_dist.items():
30        if "weight" in n:
31            prior.append(p)
32        elif "bias" in n:
33            prior[j] += p
34            prior[j] /= 2
35            prior[j] = (1-prior[j])**2
36            j += 1
37
38    return prior
39
40  def collect_grad(model, x, y, params):
41    model.zero_grad()
42    out = model(x)
43    loss = ce_loss(out, y)
44    loss.backward()
45
46    grad = {}
47    for n, p in params.items():
48      grad[n].data = p.data
49
50    return grad
```

Listing 1: PyTorch-friendly pseudo code for obtaining prior

## B APPENDIX: EXPERIMENTAL RESULTS

### B.1 ABLATION STUDIES

**Channel-wise** $\alpha$**.** In Table 8, we compared different granularity levels of interpolating weight $\alpha$, *i.e.,* channel-wise, layer-wise, and a constant value with CIFAR-10-C and backbone WRN-40. We observed that channel-wise optimized $\alpha$ shows the best performance. We take average of the channel-wisely optimized $\alpha$ (the 1st row) over channels to make a layer-wise $\alpha$ (the 2nd row), and take average over all channels and layers to make a constant value (the 3rd row). $\alpha$ of the 1st and 2nd rows are visualized in the main paper Figure 4 colored in blue and red, respectively. The constant value $\alpha$ (the 3rd row) is 0.3988. We also optimized $\alpha$ layer-wisely (the 4th row).

Table 8: **Ablation study on granularity level of** $\alpha$

| # | Method | Test batch size | | | | | | Avg. |
|---|---|---|---|---|---|---|---|---|
| | | 200 | 64 | 16 | 4 | 2 | 1 | |
| 1 | **Channel-wise (Optimized)** | **11.67** | **11.80** | **12.13** | **13.93** | **15.83** | **17.99** | **13.89** |
| 2 | Layer-wise (Channel mean) | 12.75 | 12.84 | 13.16 | 14.66 | 16.40 | 18.82 | 14.77 |
| 3 | Constant (Mean) | 12.07 | 12.21 | 13.05 | 16.72 | 20.04 | 21.26 | 15.89 |
| 4 | Layer-wise (Optimized) | 13.11 | 13.21 | 13.51 | 14.84 | 16.46 | 18.62 | 14.96 |

**MSE loss strength** $\lambda$. We empirically set the MSE loss strength $\lambda$ in Eq. 6 as 1 through the ablation study using CIFAR-10-C and WRN-40 (Table 9). Using the MSE regularizer prevents the learnable $\alpha$ from moving too far from the prior, thus letting $\alpha$ follow our intuition, i.e., putting smaller importance on the test batch statistics if the layer or channel is invariant to domain shifts. However, with too strong regularization ($\lambda$=10.0), the overall error rates are high, which means the $\alpha$ needs to be sufficiently optimized. On the other hand, with too small regularization, the $\alpha$ may overfit to the training batch size (B=200) and lose the generalizability to the smaller batch size.

Table 9: **MSE loss strength** $\lambda$

| $\lambda$ | Test batch size | | | | | | Avg. |
|---|---|---|---|---|---|---|---|
| | 200 | 64 | 16 | 4 | 2 | 1 | |
| 0.0 | 11.73 | 11.82 | 12.23 | 14.18 | 16.41 | 19.27 | 14.27 |
| 0.1 | 11.65 | 11.91 | 12.09 | 13.84 | 15.71 | 18.24 | 13.91 |
| **1.0** | 11.67 | 11.80 | 12.13 | 13.93 | 15.83 | 17.99 | 13.89 |
| 10.0 | 12.45 | 12.63 | 12.82 | 14.55 | 16.32 | 18.56 | 14.56 |

### B.2 MORE COMPARISONS ON CIFAR10-C

**Constant** $\alpha$. Table 10 shows results of simple baseline for normalization-based methods where the $\alpha$ is a constant value ranging from 0 to 1. $\alpha = 0$ is identical to CBN (Ioffe & Szegedy, 2015), and $\alpha = 1$ is identical to TBN (Nado et al., 2020). We observe that the lower $\alpha$, i.e., using less test batch statistics, shows better performance for small test batch sizes. This observation is analogous to the finding of the previous work (Schneider et al., 2020).

Table 10: **Constant** $\alpha$. Error rate ($\downarrow$) averaged over 15 corruptions of CIFAR-10-C (WRN-40).

| $\alpha$ | Test batch size | | | | | | Avg. |
|---|---|---|---|---|---|---|---|
| | 200 | 64 | 16 | 4 | 2 | 1 | |
| 0 | 18.26 | 18.39 | 18.26 | 18.26 | 18.25 | 18.25 | 18.28 |
| 0.1 | 13.95 | 14.1 | 14.05 | 14.65 | **15.14** | **15.45** | 14.56 |
| 0.2 | 12.46 | 12.66 | 12.89 | **14.30** | 15.53 | 15.64 | **13.91** |
| 0.3 | **12.05** | **12.29** | **12.72** | 15.18 | 17.35 | 17.42 | 14.50 |
| 0.4 | 12.13 | 12.41 | 13.12 | 16.69 | 19.81 | 20.51 | 15.78 |
| 0.5 | 12.42 | 12.78 | 13.73 | 18.32 | 22.52 | 24.88 | 17.44 |
| 0.6 | 12.88 | 13.32 | 14.48 | 20.02 | 25.17 | 31.97 | 19.64 |
| 0.7 | 13.37 | 13.9 | 15.23 | 21.75 | 27.91 | 46.65 | 23.14 |
| 0.8 | 13.82 | 14.37 | 15.94 | 23.44 | 30.59 | 77.15 | 29.22 |
| 0.9 | 14.18 | 14.8 | 16.58 | 24.94 | 33.12 | 89.81 | 32.24 |
| 1 | 14.50 | 15.15 | 17.10 | 26.29 | 35.67 | 90.00 | 33.12 |

### B.3 VARIANTS OF TTN FOR SMALL TEST BATCH SIZE

**Online TTN.** TTN interpolating weight $\alpha$ can also be adapted during test time. Table 11 shows the results of the TTN online version, which further optimizes the post-trained alpha using the entropy minimization and mean-squared error (MSE) loss between the updated alpha and the initial post-trained alpha. We followed entropy minimization loss used in TENT (Wang et al., 2020), and the MSE loss can be written as $\mathcal{L}_{\text{MSE}} = \|\alpha - \alpha_0\|^2$, where $\alpha_0$ is the post-trained $\alpha$. We set the learning rate as 1e-2, 2.5e-3, 5e-4, 1e-4, 5e-5, and 2.5e-5 by linearly decreasing according to the test batch size of 200, 64, 16, 4, 2, and 1 (Goyal et al., 2017). The online TTN shows improvements compared to the offline TTN in all three evaluation scenarios (single, continuously changing, and mixed).

**Scaled TTN.** Following the observation from Table 10, we lowered the interpolating weight by multiplying a constant scale value, ranging $(0, 1)$, to the optimized TTN $\alpha$. In Table 11, we empirically set the scale value as 0.4.

**Dynamic training batch size.** We observe that using the dynamic batch size in the post-training stage also improves the performance for small test batch sizes (2 or 1), while slightly deteriorating the performance for large test batch sizes (200 or 64). We randomly sampled training batch size from the range of $[4, 200]$ for each iteration. Other hyperparameters are kept as the same.

Table 11: **TTN variants.** Error rate ($\downarrow$) averaged over 15 corruptions of CIFAR-10-C (WRN-40).

| Method | Eval. setting | Test batch size | | | | | | Avg. |
|---|---|---|---|---|---|---|---|---|
| | | 200 | 64 | 16 | 4 | 2 | 1 | |
| **TTN (offline, default)** | Single & Cont. | 11.67 | 11.80 | 12.13 | 13.93 | 15.83 | 17.99 | 13.89 |
| **TTN (offline, default)** | Mixed | 12.16 | 12.19 | 12.34 | 13.96 | 15.55 | 17.83 | 14.00 |
| TTN (online) | Single | 11.39 | 11.64 | 11.97 | 13.70 | 15.41 | 17.49 | 13.60 |
| TTN (online) | Cont. | 11.67 | 11.96 | 12.15 | 13.90 | 15.67 | 17.72 | 13.85 |
| TTN (online) | Mixed | 12.04 | 12.04 | 12.06 | 13.90 | 15.46 | 17.62 | 13.85 |
| TTN (scaled) | Single & Cont. | 13.20 | 13.38 | 13.35 | 13.88 | 14.54 | 15.17 | 13.92 |
| TTN (scaled) | Mixed | 13.17 | 13.05 | 13.17 | 13.74 | 14.36 | 15.09 | 13.76 |
| TTN (dynamic train batch size) | Single & Cont. | 11.82 | 12.04 | 12.17 | 13.60 | 15.13 | 17.22 | 13.66 |
| TTN (dynamic train batch size) | Mixed | 12.12 | 12.01 | 11.91 | 13.43 | 14.76 | 17.23 | 13.58 |

## B.4 STUDY ON AUGMENTATION TYPE

**TTN is robust to the data augmentation.** We used data augmentation in the post-training phase to simulate domain shifts. The rationale for the simulation is to expose the model to different input domains. Especially when obtaining the prior, we compare the outputs from shifted domains with the clean (original) domain in order to analyze which part of the model is affected by the domain discrepancy. Therefore, changing the domain itself is what matters, not *which* domain the input is shifted to. We demonstrated this by conducting ablation studies by varying the augmentation type.

We analyzed various augmentation types when obtaining prior and when optimizing alpha and the results are shown in Figure 5 (a) and (b), respectively. We use a true corruption, *i.e.,* one of 15 corruption types in the corruption benchmark, as an augmentation type in the post-training phase to analyze how TTN works if the augmentation type and test corruption type are misaligned or perfectly aligned. Specifically, we used the true corruption when obtaining the prior while keeping the augmentation choices when optimizing alpha as described in the appendix A.1, and vice versa. Expectedly, the diagonal elements, i.e., where the same corruption type is used both for in post-training and in test time, tend to show the lowest error rate in Figure 5(b). The row-wise standard deviation is lower than 1 in most cases and even lower than 0.5 in Figure 5(a), which means the prior is invariant to the augmentation choice. Moreover, we observe that the average error rate over all elements, 11.74% in ablation for obtaining prior and 11.67% in ablation for optimizing alpha, is almost as same as TTN result 11.67% (See Table 14 and Figure 5). Moreover, we conducted an ablation study on the choice of the augmentation type using CIFAR-10-C and WRN-40 (Table12). We observe that obtaining prior and optimizing alpha steps are robust to the augmentation types.

Table 12: **Ablation study on augmentation choice.** From left to right one augmentation type is added at a time. Default setting, which we used in all experiments, is colored in gray.

| **Prior** augmentation type | color jitter | + grayscale | **+ invert** | + gaussian blur | + horizontal flip |
|---|---|---|---|---|---|
| | 12.03 | 11.83 | 11.67 | 11.59 | 11.58 |
| **Optimizing** $\alpha$ augmentation type | color jitter | + padding | + affine | + gaussian blur | **+ horizontal flip** |
| | 11.78 | 11.78 | 11.7 | 11.70 | 11.67 |

## B.5 RESULTS ON IMAGENET-C

Table 13 shows the experimental results using ResNet-50 (He et al., 2016) on ImageNet-C (Hendrycks & Dietterich, 2018) dataset. We emphasized the effectiveness of our proposed method by showing the significant improvement in large-scale dataset experiment. Similar to the results in CIFAR-10/100-C, TTN showed the best performance compared to normalization-based methods (TBN (Nado et al., 2020), AdaptiveBN Schneider et al. (2020), and $\alpha$-BN (You et al., 2021)) and improved TTA performance when it is applied to optimization-based methods (TENT (Wang et al., 2020) and SWR (Choi et al., 2022)).

During post-training, we used LR of 2.5e-4 and batch size of 64. In test time, we used the learning rate of 2.5e-4 for TENT following the setting from the original paper, and used 5e-6 for TENT+TTN. For SWR and SWR+TTN, we used learning rate of 2.5e-4 for B=64, and 2.5e-6 for B=16, 4, 2, and 1. We had to carefully tune the learning rate especially for SWR, since the method updates the

entire model parameters in an unsupervised manner and hence is very sensitive to the learning rate when the test batch size becomes small. Other hyperparameters and details are kept same (See the appendix A.1 for more details). Moreover, to avoid rapid accumulation, we stored gradients for sufficiently large test samples and then updated the model parameters (for example, we conducted adaptation in every 64 iterations in the case of batch size of 1) in both TENT and SWR.

Table 13: **Single domain adaptation on ImageNet-C (ResNet-50).** Error rate ($\downarrow$) averaged over 15 corruption types with severity level 5 is reported for each test batch size.

| | Method | Test batch size | | | | | Avg. Error |
|---|---|---|---|---|---|---|---|
| | | 64 | 16 | 4 | 2 | 1 | |
| | Source (CBN) | 93.34 | 93.34 | 93.34 | 93.34 | 93.34 | 93.34 |
| Norm | TBN | 74.24 | 76.81 | 85.74 | 95.35 | 99.86 | 86.40 |
| | AdaptiveBN | 77.86 | 81.47 | 86.71 | 90.15 | 91.11 | 85.46 |
| | $\alpha$-BN | 86.06 | 86.32 | 87.16 | 88.33 | 90.45 | 87.66 |
| | **Ours (TTN)** | **72.21** | **73.18** | **76.98** | **81.52** | **88.49** | **78.48** |
| Optim. | TENT | 66.56 | 72.61 | 93.37 | 99.46 | 99.90 | 86.41 |
| | **+Ours (TTN)** | 71.42 | **72.45** | **76.66** | **81.89** | **91.00** | **78.68** |
| | SWR | 64.41 | 74.19 | 84.30 | 93.05 | 99.86 | 83.16 |
| | **+Ours (TTN)** | **55.68** | **69.25** | **78.48** | **86.37** | **94.08** | **76.77** |

## B.6 ERROR RATES OF EACH CORRUPTION

In Table 14, we show the error rates of TTN for each corruption of CIFAR-10-C using the WRN-40 backbone. The averaged results over the corruptions are in Table 1.

Table 14: **Results of each corruption (CIFAR-10-C)**

| batch size | gauss. | shot | impulse | defocus | glass | motion | zoom | snow | frost | fog | brightness | contrast | elastic | pixel. | jpeg. | Avg. |
|---|---|---|---|---|---|---|---|---|---|---|---|---|---|---|---|---|
| 200 | 14.81 | 12.78 | 17.32 | 7.37 | 17.87 | 8.51 | 7.23 | 10.29 | 9.88 | 11.29 | 6.06 | 8.36 | 13.42 | 14.89 | 14.94 | 11.67 |
| 64 | 14.81 | 12.81 | 17.42 | 7.41 | 18.21 | 8.66 | 7.41 | 10.44 | 9.93 | 11.63 | 6.11 | 8.35 | 13.59 | 15.18 | 15.02 | 11.80 |
| 16 | 15.23 | 13.00 | 17.98 | 7.71 | 18.46 | 8.95 | 7.68 | 10.85 | 10.17 | 12.21 | 6.25 | 8.54 | 13.95 | 15.67 | 15.34 | 12.13 |
| 4 | 17.03 | 15.29 | 19.75 | 9.26 | 20.93 | 10.01 | 9.62 | 12.58 | 11.85 | 13.65 | 7.69 | 9.25 | 16.21 | 18.08 | 17.70 | 13.93 |
| 2 | 19.21 | 16.80 | 21.86 | 10.70 | 23.74 | 11.39 | 11.92 | 14.00 | 13.39 | 15.38 | 8.95 | 10.13 | 18.92 | 20.68 | 20.40 | 15.83 |
| 1 | 22.03 | 19.97 | 25.24 | 12.41 | 26.99 | 12.22 | 14.39 | 14.73 | 14.60 | 17.27 | 10.16 | 9.51 | 22.10 | 24.12 | 24.09 | 17.99 |

## B.7 SEGMENTATION RESULTS

For more comparisons, we add segmentation results for TBN and TTN using test batch size (B) of 4 and 8 (Table 15). The results demonstrate that TTN is robust to the test batch sizes. In other words, the performance difference across the test batch size is small when using TTN (TTN with B=2, 4, and 8). The results are averaged over 3 runs (i.e., using 3 independently optimized $\alpha$), and the standard deviation is denoted with a $\pm$ sign. We omitted the standard deviation for TBN, which is 0.0 for every result since no optimization is required. Since the backbone network is trained with a train batch size of 8, we assume that test batch statistics estimated from the test batch with B=8 are sufficiently reliable. Accordingly, TBN with B=8 shows compatible results. However, when B becomes small (i.e., in a more practical scenario), problematic test batch statistics are estimated, and thus TBN suffers from the performance drop while TTN keeps showing robust performance. It is worth noting that TTN outperforms TBN by 3.77% in average accuracy when B=2, i.e., in the most practical evaluation setting, and by 2.54% and 1.99% for B=4 and 8, respectively.

Table 15: **Adaptation on DG benchmarks in semantic segmentation.** mIoU($\uparrow$) on four unseen domains with test batch size of 2, 4, and 8 using ResNet-50 based DeepLabV3+ as a backbone.

| Method (Cityscapes$\rightarrow$) | | BDD-100K | Mapillary | GTAV | SYNTHIA | Cityscapes |
|---|---|---|---|---|---|---|
| Norm | TBN (B=2) | 43.12 | 47.61 | 42.51 | 25.71 | 72.94 |
| | **Ours (TTN)** (B=2) | **47.40**($\pm$ 0.02) | **56.88**($\pm$ 0.04) | **44.69**($\pm$ 0.03) | **26.68**($\pm$ 0.01) | **75.09**($\pm$ 0.01) |
| | TBN (B=4) | 45.64 | 49.17 | 44.26 | 25.96 | 74.29 |
| | **Ours (TTN)** (B=4) | **47.72**($\pm$ 0.01) | **57.11**($\pm$ 0.01) | **45.08**($\pm$ 0.02) | **26.52**($\pm$ 0.01) | **75.56**($\pm$ 0.01) |
| | TBN (B=8) | 46.42 | 49.38 | 44.81 | 25.97 | 75.42 |
| | **Ours (TTN)** (B=8) | **47.25**($\pm$ 0.02) | **57.28**($\pm$ 0.02) | **45.13**($\pm$ 0.03) | **26.45**($\pm$ 0.01) | **75.82**($\pm$ 0.01) |

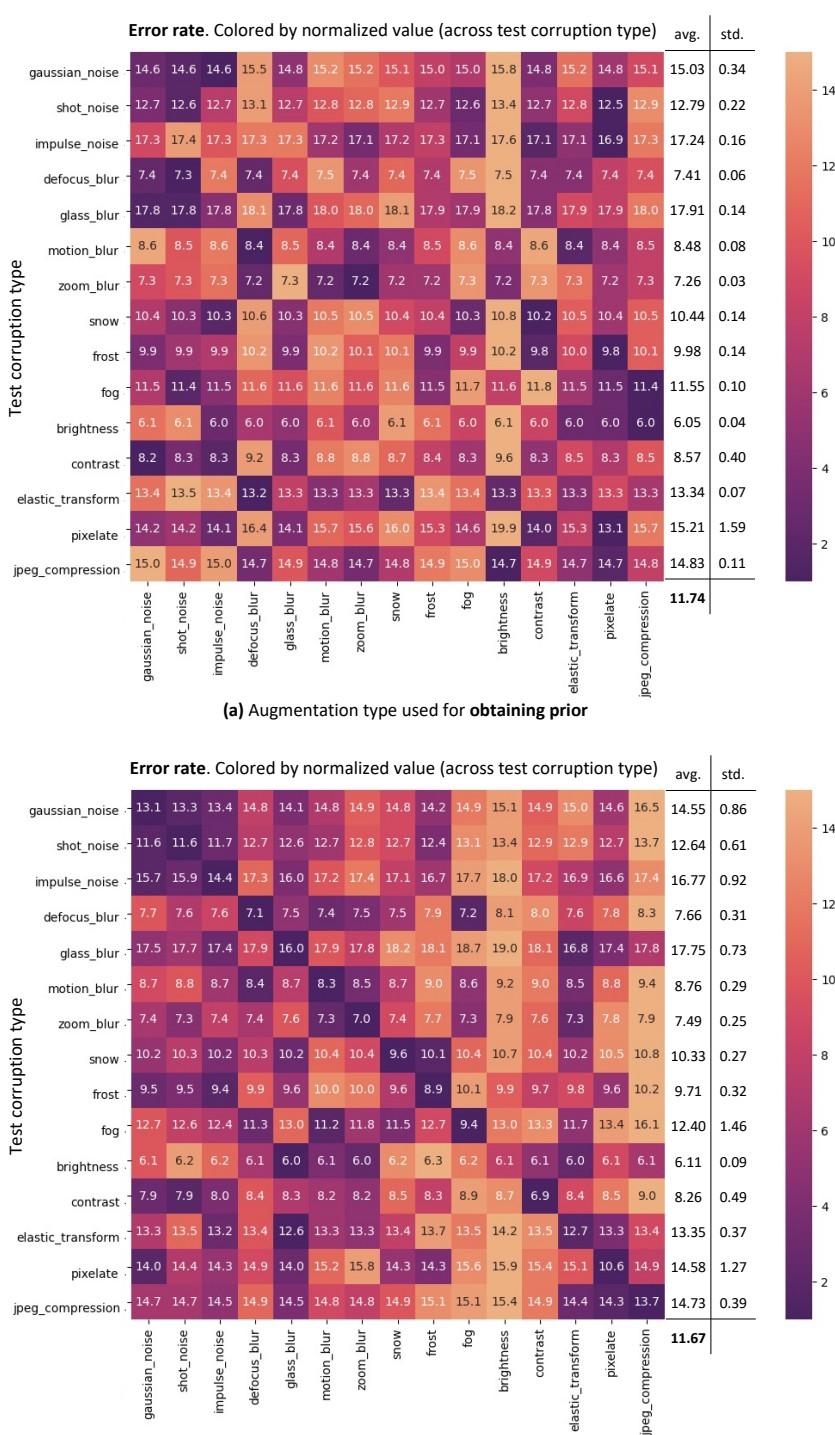

Figure 5: **True test corruption as augmentation.** Each column represents the augmentation type used **(a)** when obtaining prior or **(b)** when optimizing $\alpha$. Each row represents the test corruption type. The error rate ($\downarrow$) of CIFAR-10-C with severity level 5 and test batch size 200 using WRN-40 is annotated in each element. Element $(i, j)$ represents the error rate when $j$-th corruption type is used for augmenting the clean train data during post-training and tested on $i$-th corruption test set. The heatmap is colored by error rate normalized across the test corruption type (row-wise normalization).

