# OpenReview forum: "TTN: A Domain-Shift Aware Batch Normalization in Test-Time Adaptation"
_ICLR.cc/2023/Conference — ICLR 2023 poster_

### Official Review · Reviewer_RNFt · 2022-10-23

**Confidence:** 4
**Correctness:** 2
**Technical Novelty And Significance:** 3
**Empirical Novelty And Significance:** 3
**Recommendation:** 6

**Clarity, Quality, Novelty And Reproducibility:**

The paper is well-writen and technically sound. If the authors can address my concerns, I would raise my scoring to accept.

**Strength And Weaknesses:**

**Positive Points:**

 - The authors propose to adjust the balanced parameters between the training statistics and the test-time statistics based on the (augmented) training data.

 - The authors introduce a gradient distance score to calculate the balanced parameters $\alpha$, which is technically sound.

 - The authors consider plenty of test-time scenarios, such as various batch sizes, stationary, continuously changing, and mixed domain adaptation.

 **Negative Points:**

  - The authors obtain Prior based on the pair (clean sample x, and domain-shifted x'). And the domain-shifted x' greatly depends on the augmentation type and applying order. In this case, if the domain-shifted x' has an extensive domain gap from the test-time samples, the Prior would be estimated inaccurately, leading to poor TTA performance. Could the authors provide more discussion and explanations about this?

 - In the paragraph of "Optimize $\alpha$", the authors state "To simulate distribution shift, we use the augmented training data". Here, the authors have a strong assumption that the test-time shifted type is very similar to the augmented type. However, in my opinion, this assumption often does not hold in practice. More explanations are required.

 - I suggest the authors can provide an ablation study on the augmentation type, which is very crucial for the proposed method. I wonder if there is an augmentations combination that works well for all the corruption-shifted test-time samples? If no, the proposed method may be difficult for real-world applications.

 - Most experimental results are reported on CIFAR-C. I suggest the authors can report the main results on ImageNet-C, which will be more convincing. I think the computational and time cost on ImageNet-C is similar to that on CIFAR-C. In Table 9, I find the authors do not report the results of "Tent+TTN" and "SWR+TTN". But I think these two are crucial results.

**Summary Of The Paper:**

**Summary:**

This paper presents a new test-time normalization (TTN) method that combines the training statistics and the test-time statistics via the importance between them. To obtain the Prior, the authors first augment the training samples as domain-shift ones, and then calculate the importance via the gradient distance score between the clean samples and domain-shift ones. The experimental results on CIFAR-C and ImageNet-C demonstrate the effectiveness of the proposed method.

**Summary Of The Review:**

I vote for reject since the proposed method has a strong assumption on the test samples, which may not be hold in practice.

---

> ### Author Response · Authors · 2022-11-16
> **Response to Reviewer RNFt (Q4.1- 4.4)**
>
> We appreciate the thorough and constructive feedback. To address the concern mainly about augmentation type, we conducted several experiments. Moreover, we added more comparisons to ImageNet-C. We hope our responses address all the concerns.
>
> **Q4.1: The authors obtain Prior based on the pair (clean sample x, and domain-shifted x'). And the domain-shifted x' greatly depends on the augmentation type and applying order. In this case, if the domain-shifted x' has an extensive domain gap from the test-time samples, the Prior would be estimated inaccurately, leading to poor TTA performance. Could the authors provide more discussion and explanations about this?**
>
> **Q4.2: In the paragraph of "Optimize ", the authors state "To simulate distribution shift, we use the augmented training data". Here, the authors have a strong assumption that the test-time shifted type is very similar to the augmented type. However, in my opinion, this assumption often does not hold in practice. More explanations are required.**
>
> **Q4.3: I suggest the authors can provide an ablation study on the augmentation type, which is very crucial for the proposed method. I wonder if there is an augmentations combination that works well for all the corruption-shifted test-time samples? If no, the proposed method may be difficult for real-world applications.**
>
> **A4.1, A4.2, A4.3**: To show that our method is robust and invariant to the choice of the augmentation type, we analyzed various augmentation types when obtaining prior (step 1)  and when optimizing alpha (step 2). We conducted two experiments and we added the results in **Appendix B.4**.
>
> First, we use a true corruption (i.e., one of 15 corruption types in the corruption benchmark) as an augmentation type in the post-training phase to analyze how TTN works if the augmentation type and test corruption type are misaligned or perfectly aligned. In Figure 5, we used the true corruption when obtaining the prior while keeping the augmentation choices when optimizing alpha as we used in the paper (details are provided in **Appendix A.1**), and vice versa. Expectedly, the diagonal elements, i.e., where the same corruption type is used both in post-training and in test time, tend to show the lowest error rate in ablation for step 2. The row-wise standard deviation is lower than 1 in most cases and even lower than 0.5 in ablation for step  1, which means the prior is invariant to the augmentation choice. Moreover, we observe that the average error rate over all elements, 11.74 in ablation for obtaining prior and 11.67 in ablation for optimizing alpha, is almost as same as TTN result 11.67 (please see **Table 14** and **Figure 5**).
>
> Secondly, we conduct an ablation study on each augmentation type for obtaining prior (step 1) and optimizing alpha (step 2). The default choice used in the paper is highlighted in gray color (**Table 12**). The ablation also showed that obtaining prior and optimizing alpha steps are robust to the augmentation type choices.
>
> **Q4.4: Most experimental results are reported on CIFAR-C. I suggest the authors can report the main results on ImageNet-C, which will be more convincing. I think the computational and time cost on ImageNet-C is similar to that on CIFAR-C. In Table 9, I find the authors do not report the results of "Tent+TTN" and "SWR+TTN". But I think these two are crucial results.**
>
> **A4.4**: We observed that TTN shows promising result in ImageNet-C when used with optimization-based method, SWR. The error rate of SWR using test batch size (B) of 64 is 64.41% and that of SWR+TTN is 55.68% (we set B=60 due to  GPU memory issue). We will do our best to add more comparisons on ImageNet-C in the remaining time.

---

> > ### Comment · Reviewer_RNFt · 2022-11-17
> > **Response to authors**
> >
> > The authors provide extensive results in Figure 5 and Table 12. The results address most of my concerns.
> >
> > However, I still think the results in CIFAR-10 is not very convincing. I have found some method works well on a small and toy dataset but can not be applied to the large-scale and more practical one.
> >
> > In addition, In A4.4, does the revised paper include this new result?

---

> > > ### Author Response · Authors · 2022-11-17
> > > **Response to Reviewer RNFt**
> > >
> > > We have updated the results and added more comparisons to ImageNet-C (please refer to the [updated response for A4.4](https://openreview.net/forum?id=EQfeudmWLQ&noteId=z9oIdjA-skV)).
> > >
> > > Meanwhile, we hope the results of the semantic segmentation task (please refer to **Section 3.3** and **Table 6** for the results and our responses [A1.3.1](https://openreview.net/forum?id=EQfeudmWLQ&noteId=1f_6NbEVz3l) and [A2.5](https://openreview.net/forum?id=EQfeudmWLQ&noteId=Ny2Io7namGq) for the related discussions), where we use large-scale real-world (Citypscapes, BDD-100K, and Mapiliary) and synthetic (GTAV and SYNTHIA) datasets, help the reviewer address the concerns. In the experiment, we demonstrated that our proposed method is broadly applicable to different tasks and realistic datasets.

---

> > > > ### Comment · Reviewer_RNFt · 2022-12-05
> > > > **Response**
> > > >
> > > > Thanks for the reponses. All of my concerns have been well addressed. I would raise my scoring to weak accept. I think it is amazing that Piror with arbitrary augmentation type in the post-training phase can handle various corruptions in the test phase. This is an important insight for the community. It would be better for the authors to provide more disscussions about why this work in the revised version.

---

> > > ### Author Response · Authors · 2022-11-18
> > > **Response to Reviewer RNFt (A4.4 updated)**
> > >
> > >
> > > We have updated the results in ImageNet-C in **Table 13**. Additional to TENT, TENT+TTN, SWR, and SWR+TTN, we added AdaptiveBN (Schenider et. al, 2020) as a strong baseline for normalization-based methods (please refer to our response [A2.3](https://openreview.net/forum?id=EQfeudmWLQ&noteId=Ny2Io7namGq) for results of CIFAR-10/100-C). We emphasized the effectiveness of our proposed method by showing the significant improvement in large-scale dataset experiments. Similar to the results in CIFAR-10/100-C, TTN showed the best performance compared to normalization-based methods and improved TTA performance when it is applied to optimization-based methods.
> > >
> > > We used the learning rate of 2.5e-4 for TENT following the setting from the original paper, and we used 5e-6 for TENT+TTN.
> > > For SWR and SWR+TTN, we used the learning rate of 2.5e-4 for B=64, and 2.5e-6 for B=16, 4, 2, and 1. We had to tune the learning rate carefully for SWR. Since SWR updates the entire model parameters in an unsupervised manner, it is very sensitive to the learning rate when the test batch size becomes small, especially when using a challenging dataset. For example for B=16, (SWR, SWR+TTN) showed error rates of (99.49, 99.79), (87.21, 86.27), and (74.19, 69.25) for LR of 2.5e-4, 2.5e-5, and 2.5e-6, respectively. Accordingly, we set LR as 2.5e-6 for B smaller than 64.
> > > Moreover, to avoid rapid accumulation, we stored gradients for sufficiently large test samples and then updated the model parameters (for example, we conducted adaptation in every 64 iterations in the case of a batch size of 1) in both TENT and SWR, as we did in the other experiments.
> > >
> > > | Method |64 | 16 | 4 | 2 | 1 | Avg
> > > |---|---|---|---|---|---|---|
> > > | Source (CBN) | 93.34 | 93.34 | 93.34 | 93.34 | 93.34 | 93.34
> > > | TBN | 74.24 | 76.81 | 85.74 | 95.35 | 99.86 | 86.40
> > > | AdaptiveBN | 77.86 | 81.47 | 86.71 | 90.15 | 91.11 | 85.46
> > > | alpha-BN| 86.06| 86.32 | 87.16 | 88.33 | 90.45| 87.66
> > > | Ours (TTN) | **72.21** | **73.18** | **76.98** | **81.52** | **88.49** | **78.48**
> > > | TENT | **66.56** | 72.61|	93.37|	99.46| 99.90 | 86.41
> > > | TENT+**Ours(TTN)** | 71.42 | **72.45** | **76.66** | **81.89** | **91.00** | **78.68**
> > > | SWR | 64.41 | 74.19 | 84.30 | 93.05 | 99.86 | 83.16
> > > | SWR+**Ours(TTN)** | **55.68** | **69.25** | **78.48**| **86.37** | **94.08** | **76.77**
> > >
> > > Error rates along the different learning rate for SWR when B=16 are as follows:
> > >
> > > | Method | LR | Error rate
> > > |---|---|---
> > > |SWR|2.5e-4| 99.49
> > > |SWR+**Ours(TTN)**|2.5e-4| 99.79
> > > |SWR|2.5e-5| 87.21
> > > |SWR+**Ours(TTN)**|2.5e-5| 86.27
> > > |SWR|2.5e-6| 74.19
> > > |SWR+**Ours(TTN)**|2.5e-6| 69.25

---

> > ### Author Response · Authors · 2022-11-17
> > **Summary of experimental results (A4.1, A4.2, and A4.3)**
> >
> >
> > **A4.1, A4.2, A4.3 (continued)**: As we updated in **Figure 5**, we conducted ablation studies on the augmentation types. We report the results here once again for improved readability. Element (i, j) represents the error rate when j-th corruption type is used for augmenting the clean train data during post-training and tested on i-th corruption test set. The results are as follows:
> >
> > - Ablation for obtaining prior
> >
> > |   |gauss|	shot|	impls|	defcs|	glass|	motion|	zoom|	snow|	frost|	fog	| brght|	cntrst|	elastic|	pixel|	jpeg|	avg|	std|
> > |--- |---|---|---|---|---|---|---|---|---|---|---|---|---|---|---|---|---|
> > |gauss	|14.64	|14.63	|14.57	|15.53	|14.80	|15.23	|15.20	|15.13	|14.97	|14.96	|15.79	|14.79	|15.20	|14.84	|15.11	|15.03	|0.34|
> > |shot	|12.66	|12.62	|12.73	|13.07	|12.73	|12.84	|12.85	|12.87	|12.70	|12.63	|13.41	|12.68	|12.75	|12.47	|12.90	|12.79	|0.22|
> > |impls	|17.27	|17.37	|17.28	|17.29	|17.29	|17.16	|17.14	|17.25	|17.26	|17.14	|17.65	|17.09	|17.15	|16.91	|17.29	|17.24	|0.16|
> > |defcs	|7.36	|7.30	|7.44	|7.36	|7.42	|7.48	|7.38	|7.43	|7.39	|7.48	|7.52	|7.37	|7.37	|7.41	|7.42	|7.41	|0.06|
> > |glass	|17.75	|17.75	|17.78	|18.05	|17.77	|17.98	|18.00	|18.06	|17.88	|17.85	|18.23	|17.78	|17.91	|17.89	|18.04	|17.91	|0.14|
> > |motion	|8.60	|8.53	|8.56	|8.36	|8.54	|8.45	|8.40	|8.44	|8.50	|8.57	|8.44	|8.60	|8.37	|8.44	|8.47	|8.48	|0.08|
> > |zoom	|7.27	|7.28	|7.29	|7.22	|7.32	|7.22	|7.20	|7.25	|7.25	|7.30	|7.24	|7.30	|7.29	|7.25	|7.26	|7.26	|0.03|
> > |snow	|10.37	|10.35	|10.33	|10.55	|10.35	|10.53	|10.54	|10.43	|10.39	|10.34	|10.81	|10.23	|10.49	|10.36	|10.49	|10.44	|0.14|
> > |frost	|9.86	|9.87	|9.88	|10.15	|9.86	|10.15	|10.08	|10.09	|9.86	|9.95	|10.20	|9.83	|10.03	|9.81	|10.08	|9.98	|0.14|
> > |fog	    |11.51	|11.43	|11.48	|11.58	|11.57	|11.60	|11.57	|11.60	|11.48	|11.72	|11.56	|11.75	|11.49	|11.47	|11.40	|11.55	|0.10|
> > |brght	|6.07	|6.09	|6.01	|6.04	|6.02	|6.06	|6.04	|6.11	|6.07	|6.05	|6.12	|6.05	|6.01	|6.01	|6.00	|6.05	|0.04|
> > |cntrst	|8.23	|8.28	|8.26	|9.16	|8.28	|8.79	|8.85	|8.69	|8.45	|8.33	|9.63	|8.31	|8.48	|8.34	|8.50	|8.57	|0.40|
> > |elastic	|13.40	|13.46	|13.44	|13.25	|13.33	|13.29	|13.32	|13.26	|13.44	|13.40	|13.26	|13.35	|13.29	|13.34	|13.32	|13.34	|0.07|
> > |pixel	|14.15	|14.21	|14.12	|16.45	|14.14	|15.66	|15.64	|15.99	|15.33	|14.64	|19.85	|14.02	|15.26	|13.05	|15.69	|15.21	|1.59|
> > |jpeg	|14.99	|14.91	|14.97	|14.72	|14.89	|14.76	|14.72	|14.79	|14.93	|14.96	|14.70	|14.88	|14.74	|14.71	|14.82	|14.83	|0.11|
> > |Avg. ||||||||||||||||**11.74**|
> >
> > - Ablation for optimizing alpha
> >
> > |   |gauss|	shot|	impls|	defcs|	glass|	motion|	zoom|	snow|	frost|	fog	| brght|	cntrst|	elastic|	pixel|	jpeg|	avg|	std|
> > |---|---|---|---|---|---|---|---|---|---|---|---|---|---|---|---|---|---|
> > |gauss|	13.05|	13.27|	13.38|	14.79|	14.08|	14.78|	14.94|	14.80|	14.22	|14.95	|15.05	|14.94	|15.02	|14.57	|16.46	|14.55	|0.86|
> > |shot|	11.62|	11.56|	11.72|	12.71|	12.60|	12.69|	12.77|	12.74|	12.41	|13.06	|13.43	|12.90	|12.94	|12.71	|13.68	|12.64	|0.61|
> > |impls|	15.74|	15.88|	14.40|	17.31|	16.02|	17.20|	17.40|	17.08|	16.69	|17.68	|17.96	|17.16	|16.92	|16.62	|17.42	|16.77	|0.92|
> > |defcs|	7.72|	7.56|	7.58|	7.14|	7.54|	7.40|	7.48|	7.55 |  7.88	|7.23	|8.05	|8.03	|7.58	|7.85	|8.26	|7.66	|0.31|
> > |glass|	17.46|	17.66|	17.35|	17.90|	15.99|	17.92|	17.79|	18.22|  18.12	|18.72	|19.04	|18.07	|16.78	|17.41	|17.76	|17.75	|0.73|
> > |motion|8.74|	8.76|	8.70|	8.41|	8.73|	8.27|	8.53|	8.73 |  9.02	|8.59	|9.20	|8.99	|8.53	|8.82	|9.38	|8.76	|0.29|
> > |zoom|	7.37|	7.34|	7.43|	7.37|	7.57|	7.27|	7.01|	7.44 |  7.72	|7.34	|7.90	|7.58	|7.27	|7.77	|7.92	|7.49	|0.25|
> > |snow|	10.25|	10.32|	10.24|	10.32|	10.18|	10.44|	10.43|	9.59 |	10.14	|10.44	|10.66	|10.44	|10.25	|10.53	|10.75	|10.33	|0.27|
> > |frost|	9.50|	9.54|	9.37|	9.90|	9.56|	9.99|	9.96|	9.64 |  8.95	|10.09	|9.90	|9.74	|9.76	|9.56	|10.18	|9.71	|0.32|
> > |fog|   12.70|	12.62|	12.38|	11.35|	12.96|	11.15|	11.85|	11.51|	12.66	|9.36	|13.00	|13.33	|11.65	|13.44	|16.09  |12.40  |1.46|
> > |brght|	6.14|	6.21|	6.15|	6.10|	5.98|	6.08|	5.99|	6.18 |  6.30	|6.16	|6.12	|6.09	|5.99	|6.13	|6.10	|6.11	|0.09|
> > |cntrst|7.94|	7.89|	8.02|	8.42|	8.26|	8.23|	8.16|	8.49 |  8.27	|8.86	|8.66	|6.89	|8.37	|8.52	|8.96   |8.26   |0.49|
> > |elastic|13.35|	13.47|	13.21|	13.41|	12.59|	13.32|	13.29|	13.39|	13.70	|13.48	|14.19	|13.53	|12.69	|13.28	|13.40  |13.35	|0.37|
> > |pixel|	13.99|	14.36|	14.35|	14.94|	14.03|	15.15|	15.75|	14.35|	14.33	|15.59	|15.86	|15.43	|15.09	|10.56	|14.92	|14.58	|1.27|
> > |jpeg|	14.73|	14.73|	14.52|	14.87|	14.46|	14.85|	14.84|	14.86|	15.05	|15.14	|15.40	|14.95	|14.42	|14.35	|13.74	|14.73	|0.39|
> > |Avg. ||||||||||||||||**11.67**|

---

> ### Author Response · Authors · 2022-12-05
> **Kind reminder for discussion**
>
> Dear reviewer RNFt,
>
> We are confident that your insightful comments have made our paper more solid.
>
> Following the feedback Q4.1, 4.2, and 4.3, we have conducted extensive augmentation ablations and showed that our post-training phase (i.e., obtaining the prior and optimizing the alpha) is robust to the choice of augmentation type.
> As a response to Q4.4, we have added more results in ImageNet-C, which is a large-scale dataset for common corruption, and showed that our proposed method shows the best performance compared to normalization-based methods and improves TTA performance on top of optimization-based (backpropagation-based) methods. Furthermore, our responses at A1.3.1 and A2.5 are also related to A4.4. Here, we have conducted experiments in semantic segmentation tasks using large-scale real-world datasets, including natural shifts and real-world to synthetic domain shifts. Similar to the results for ImageNet-C, we have shown that using TTN alone or adopting TTN with backpropagation-based methods improves performance compared to the baseline methods.
>
> Kindly let us know if you have any further comments or suggestions. We will do our best to address them.
>
> Best,
>
> Paper 4623 authors

---

> ### Author Response · Authors · 2022-12-05
> **Thank you for raising the score!**
>
> We deeply appreciate raising the score and providing positive support.
>
> We are pleased to receive affirmative comments on our ablation results mentioning them as *amazing*. Following your constructive feedback, we will revise our manuscript emphasizing the robustness against the choice of augmentation types and will do our best to provide sharpened insights by adding more discussions.
>
> With all due respect, we would appreciate it if you could also consider updating the “Summary Of The Review” in the official review in accordance with the raised score.

---

### Official Review · Reviewer_EgJB · 2022-10-23

**Confidence:** 3
**Correctness:** 4
**Technical Novelty And Significance:** 2
**Empirical Novelty And Significance:** 3
**Recommendation:** 6

**Clarity, Quality, Novelty And Reproducibility:**

The clarity is clear, the quality is overall good, and the novelty is somewhat incremental. I believe the experiments can be somewhat reproduced based on the descriptions of this paper.

**Strength And Weaknesses:**


**Strengths:**

1.The motivation is clear, and the effectiveness of the proposed method makes sense, based on the previous works.

2. This paper provides a comprehensive discussion to the related work, as to me.

3. This paper conducts comprehensive experiments, and the experiments demonstrate the proposed method is empirically successful for test-time adaptation, particularly in the scenes for small batch size adaptation.

**Weaknesses:**

1.The technical novelty and contributions are incremental. Test-time adaptation using BN is recently widely investigated in previous works, as stated in the related work. Furthermore, the idea of combining the statics of BN from the source domain and targeted domain is not new from previous work. Besides, the method (calculate the domain shift sensitivity) to obtain the prior of $\alpha$ is from Choi et al. (2022), the optimization (for the interpolating weight $\alpha$) using cross entropy is also shown in several previous work (even though not specific to the interpolating weight $\alpha$). I donot well recognize new point for this paper.

2. It should provide the code (including script with hyper-parameters), since this paper conducts comprehensive experiments and the configs of these experiments are complicated. Even though this paper provides details for setup of experiments in the supplementary material, I believe it is not easy to reproduce the results based on these descriptions.


**Summary Of The Paper:**

This paper proposes a test-time normalization (TTN) method that combines the normalization statistics of source domain and target domain (during test), by adjusting the importance between them according to the domain-shift sensitivity of each BN layer. The motivation of the proposed method is clear, and the comprehensive experiments conducted in this paper demonstrate the effectiveness of the proposed method for test-time adaptation, particularly in the scenes for small batch size adaptation.

**Summary Of The Review:**

An empirical successful method for test-time adaptation but with incremental technical novelty and contribution. I am slightly positive to this paper, considering its empirical effectiveness, but not confident to accept it.

---

> ### Author Response · Authors · 2022-11-16
> **Response to Reviewer EgJB (Q3.1, Q3.2)**
>
> We appreciate the thorough feedback and positive support. We hope our responses address the concerns regarding to the novelty and code release.
>
> **Q3.1: The technical novelty and contributions are incremental. Test-time adaptation using BN is recently widely investigated in previous works, as stated in the related work. Furthermore, the idea of combining the statics of BN from the source domain and targeted domain is not new from previous work. Besides, the method (calculate the domain shift sensitivity) to obtain the prior of  is from Choi et al. (2022), the optimization (for the interpolating weight ) using cross entropy is also shown in several previous work (even though not specific to the interpolating weight ). I donot well recognize new point for this paper.**
>
> **A3.1**: We argue that the novelty of this paper lies in diversifying the interpolating weight for each layer and channel based on the model’s domain-shift sensitivity. We measure a model’s domain-shift sensitivity using the gradient distance score, which was introduced by Choi et al. (2022) to regularize the model parameter update, and we propose to use the score for standardization statistics calibration. In detail, we compute the gradient distance score of affine parameters of BN layers to obtain the prior knowledge of alpha, which implies our intuition, and we optimize the alpha based on the prior.
>
> In addition, since the interpolating weight alpha is fixed in the test time, TTN does not require any additional computation other than one forward propagation for the prediction. Even without any further backpropagation, TTN successfully adapts to a wide range of test-time domain distributions and achieves impressive TTA performance in various evaluation settings. When we adopt TTN with optimization-based methods, we achieve state-of-the-art performance. Moreover, by visualizing the optimized alpha (**Figure 4**) and explaining why using domain-shift sensitivity level for calibrating the BN statistics is helpful for improving TTA performance (**Section 3.5**), we emphasized the importance of diversifying alpha value across layers and channels (Please see **Table 8** for more details).
>
> **Q3.2: It should provide the code (including script with hyper-parameters), since this paper conducts comprehensive experiments and the configs of these experiments are complicated. Even though this paper provides details for setup of experiments in the supplementary material, I believe it is not easy to reproduce the results based on these descriptions.**
>
> **A3.2**: To ensure reproducibility, we added PyTorch-friendly pseudo-code in **Appendix A.3**.  Unfortunately, we cannot release the code because of a confidential issue at this moment. However, we will do our best to make the code publicly available in the remaining time.

---

> ### Author Response · Authors · 2022-12-05
> **Kind reminder for discussion**
>
> Dear reviewer EgJB,
>
> We are thankful for your valuable feedback, which improves our paper’s completeness.
>
> We have already provided detailed responses to the comments, which can be summarized as follows:
>
> - We have summarized the strength of our paper in aspects of novelty, efficiency, and applicability as a response to Q3.1. We will clarify our contributions in the final version of the manuscript.
> - Moreover, we have added detailed PyTorch-friendly pseudo-code to ensure reproducibility (Q3.1).
>
> Please let us know if you have any further comments or feedback. We will do our best to address them.
>
> Best,
>
> Paper 4623 authors

---

### Official Review · Reviewer_4Hgx · 2022-10-24

**Confidence:** 4
**Correctness:** 3
**Technical Novelty And Significance:** 3
**Empirical Novelty And Significance:** 3
**Recommendation:** 8

**Clarity, Quality, Novelty And Reproducibility:**

- Clarity: The paper is easy to read and understand.
- Quality: Good.
- Novelty: Interesting and somewhat novel idea.

**Strength And Weaknesses:**

**Strength**
- The proposed method is simple yet effective
- The proposed method can handle various different test settings and achieve promising performance
- The authors conduct detailed ablation studies to help the readers to better understand the method

**Weakness**
1. Missing references
- [A1][A2] are two recent state-of-the-art test-time adaptation methods, it would be good to add and compare with them
- The idea of learning interpolating weight is quite similar to [A4]. I know [A4] should be counted as concurrent work and the authors do not have to cite it. But I suggest the authors add and compare with it, to provide a better context to the readers in this area.
2. More comparisons
- I think (Schneider et al. 2020) is a strong baseline and should be included.
- The two state-of-the-art contrastive learning-based methods should also be compared
- Maybe add more comparisons for the semantic segmentation experiments as well. For example, [A3][A4].
3. Others.
- Which TBN method is used in Figure 1? According to my experience, at least (Schneider et al. 2020) should have a reasonable performance (not significantly worse than CBN). Please add more details here.
- The authors propose a mixed domain evaluation setting, where each batch contains a different domain. However, it is also possible that a single batch contains multiple domains of data. I wonder whether the proposed method can handle this case. [A3][A4] can both handle this and I think the authors can have some study or discussion about this setting.
- How to select the $\lambda$ in Eqn. (6) to balance the two loss terms?
- It is quite interesting to see that the proposed method actually performs worse than non-learning-based normalization methods when the batch size is very small (2 or 1). Can the authors comment on that? Will a dynamic batch size in the post-training stage improve the performance in this case?
- Why the normalization methods are not shown in Table 2 and Table 3? This looks like normalization methods cannot be used in these settings, but actually they can.
- While adding the proposed method improves upon the two optimization-based methods in Table 3, the results are even worse than applying the proposed method alone. This may weaken the proposed method and probably the authors can move it to the appendix.

**Reference**
- [A1] Liu et al. TTT++: When Does Self-Supervised Test-Time Training Fail or Thrive? NeurIPS 2021
- [A2] Chen et al. Contrastive Test-time Adaptation. CVPR 2022
- [A3] Khurana et al. SITA: Single Image Test-time Adaptation.
- [A4] Zou et al. Learning Instance-Specific Adaptation for Cross-Domain Segmentation. ECCV 2022


**Summary Of The Paper:**

This paper proposes a test-time batch normalization method to tackle domain shifts. The core idea is to learn interpolating vectors to combine running statistics and test-time batch statistics in batch normalization layers. The authors also propose to use a gradient distance score to initialize and regularize the interpolating vectors during training. The proposed method achieves promising results on several image classification and segmentation benchmarks. The authors also conduct extensive ablation studies to validate the design choices.

**Summary Of The Review:**

Overall, I think this paper tackles an interesting and important task (test-time adaptation), and the proposed method is well-motivated and effective. I tend to accept this paper. But I suggest the authors add some comparisons and discussion with the missing references I mentioned above.

-----

Post-rebuttal

After reading other reviews and the response, I decided to increase my rating.

---

> ### Author Response · Authors · 2022-11-16
> **Response to Reviewer 4Hgx (Q2.1-Q2.5)**
>
> Thank you for the constructive feedback and positive support. We hope our responses address all the concerns. The main concern was to add missing references and more comparisons and to add details. We added recommended baselines and comparisons and made experiments more thorough.
>
> **Q2.1: [A1][A2] are two recent state-of-the-art test-time adaptation methods, it would be good to add and compare with them.**
>
> **Q2.2: The idea of learning interpolating weight is quite similar to [A4]. I know [A4] should be counted as concurrent work and the authors do not have to cite it. But I suggest the authors add and compare with it, to provide a better context to the readers in this area.**
>
> **Q2.4:The two state-of-the-art contrastive learning-based methods should also be compared**
>
> **A2.1, A2.2, A2.4**: We added the recommended references in Related Work (**Section 4**).
>
> TTT++ [A1] optimizes the model parameters in the test time through contrastive learning. Therefore for TTT++, we need a backbone network, which we used WideResNet-40-2 in most experiments, pre-trained with contrastive methods. Unfortunately, the official TTT++ repository does not provide the pre-trained model checkpoint for WideResNet-40-2 architecture. For this reason, we could not provide a fair comparison with TTT++ at this moment. However, we will do our best to compare with TTT++ in the remaining time or by the camera-ready. Moreover, since TTT++ updates the model for multiple epochs in the test time (https://github.com/vita-epfl/ttt-plus-plus/issues/3), we believe this method is different from our optimization-based baselines (TENT and SWR) concerning the updating strategy and is closer to the source-free domain adaptation.
>
> AdaContrast [A2] refines the pseudo labels using nearest-neighbor soft voting and then optimizes the model parameters via contrastive learning. Since AdaContrast is evaluated in VisDA-C and DomainNet-126, it is not easy to validate the performance of the reproduced model in the corruption benchmark during the rebuttal. Similarly, we will do our best to provide a fair comparison in the remaining time or by the camera-ready.
>
> The concurrent work InstCal [A4] and our proposed method TTN share the idea to optimize the interpolating weight in BN layers using augmented source data while freezing other model parameters. However, while InstCal and TTN both use cross-entropy loss as an objective function, TTN additionally uses a regularization, which forces the interpolating weight to follow the prior, which implies the domain-shift sensitivity. We observed that using the model’s domain-shift sensitivity level for initialization and regularization shows better performance (especially for small test batch sizes) than not using them (Please refer to the ablation study result in **Table 7** for details). Moreover, when we visualize the optimized alpha (Figure 4), we can observe that the interpolating weight decreases as the layer gets deeper (i.e., the deeper layers use less test batch statistics because they are more agnostic to the input domain). The tendency is analogous to the observations of previous works (Pan et al., 2018; Wang et al., 2021; Kim et al., 2022a) that argue semantic information remains in deeper layers while domain information is handled in shallower layers. This analysis explains why using the domain-shift sensitivity level helps determine the interpolating weight.
>
> **Q2.3: I think (Schneider et al. 2020) is a strong baseline and should be included.**
>
> **Q2.5: Maybe add more comparisons for the semantic segmentation experiments as well. For example, [A3][A4].**
>
> **A2.3, A2.5**:
> We added AdaptiveBN (Schenider et al., 2020) as a baseline for normalization-based methods in **Table 1** and **Table 3**. AdaptiveBN adjusts the interpolating weight using two factors: hyperparameter N and test batch size n. We followed the suggested N, which is empirically obtained best hyperparameter from the original paper, for each n (Figure 11, ResNet architecture from Schneider et al. (2020)), i.e., N as 256, 128, 64, 32, and 16 for test batch size 200, 64, 16, 4, 2, and 1, which yields alpha as 0.44, 0.33, 0.2, 0.11, 0.06, and 0.06, respectively. Similar to our observation from **Table 10** (constant alpha result), the suggested N shows that test batch statistics should be used less when the test batch size is small.
>
> For semantic segmentation, we added more comparisons in **Table 6** using baseline methods, that we employed in image classification experiments. We also updated the implementation details in **Appendix A.1**.

---

> > ### Author Response · Authors · 2022-11-17
> > **Summary of updated experimental results (A2.3, A2.5)**
> >
> >
> > **A2.3(continued)**: As we updated in **Table 1** and **Table 3**, we added AdaptiveBN as our baseline and the results are as follows:
> >
> > |Dataset   |Method  |Eval Scenario| 200 |  64 |  16 |  4  |  2  |  1  | Avg |
> > |----------|--------|-------- |-----|-----|-----|-----|-----|-----|------
> > CIFAR-10-C|AdaptiveBN|Single\&Cont. |12.21|12.31	|12.89	|14.51	|15.79	|16.14	|13.98
> > -- |**TTN (Ours)**|Single\&Cont.|11.67	|11.80	|12.13|	13.93|	15.83	|17.99	|13.89
> > --|AdaptiveBN|Mixed         |12.62	|12.48|	12.97	|14.59	|15.74|	16.02|	14.07
> > -- |**TTN (Ours)**|Mixed    |12.16|	12.19|12.34|13.96|15.55|17.83|14.00
> > CIFAR-100-C|AdaptiveBN|Single\&Cont.|36.56|36.85	|38.19	|41.18	|43.26	|44.01	|40.01
> > --|**TTN (Ours)**|Single\&Cont.|35.58|35.84|36.73|41.08|46.67|57.71|42.27
> > --|AdaptiveBN|Mixed        |36.88	|36.86|	38.49	|41.43	|43.38|	44.31|	40.225
> > --|**TTN (Ours)**|Mixed    |36.24|36.23|36.85|41.01|45.85|55.52|41.95
> >
> > **A2.5(continued)**: As we updated in **Table 6**, we added more comparisons for semantic segmentation and the results are as follows:
> >
> > |  Method | BDD-100K  | Mapiliary  | GTAV  | SYNTHIA  | Cityscapes (Source)
> > |---|---|---|---|---|---|
> > | TBN | 43.12 | 47.61 | 42.51 | 25.71 |  72.94
> > | **Ours (TTN)** | 47.40 |56.92| 44.71| 26.68 |75.09
> > |TENT	    |43.3	|47.8	|43.57	|25.92	|72.93
> > |TENT+**Ours (TTN)**	|47.89	|57.84	|46.18	|27.29	|75.04
> > |SWR	    |43.4	|47.95	|42.88	|25.97	|72.93
> > |SWR+**Ours (TTN)**	|48.85	|59.09	|46.71	|29.16	|74.89

---

> ### Author Response · Authors · 2022-11-16
> **Response to Reviewer 4Hgx (Q2.6-2.9)**
>
> **Q2.6: Which TBN method is used in Figure 1? According to my experience, at least (Schneider et al. 2020) should have a reasonable performance (not significantly worse than CBN). Please add more details here.**
>
> **A2.6**: We used prediction-time batch normalization (Nado et al., 2020), the same as setting the interpolating weight as 1 in (i.e., use only test batch statistics), and denoted as TBN. We revised the paper by clearly denoting the citation and adding details. Since two strong TTA baselines, TENT and SWR, optimize model parameters on top of the prediction-time batch normalization, TENT and SWR suffer from significant performance drop for small test batch size as prediction-time batch normalization does. To show this limitation, we compared the normalization method from Nado et al., 2020 in **Figure 1**.
>
> **Q2.7: The authors propose a mixed domain evaluation setting, where each batch contains a different domain. However, it is also possible that a single batch contains multiple domains of data. I wonder whether the proposed method can handle this case. [A3][A4] can both handle this and I think the authors can have some study or discussion about this setting.**
>
> **A2.7**: As the reviewer pointed out, we modified the mixed domain adaptation setting (**Table 3**) to a single batch containing multiple domains instead of each batch containing a different domain. In detail, we randomly chose a corruption type among 15 corruptions for each of the test instances.
>
> The mixed domain adaptation setting is the most challenging but realistic scenario among the three evaluation settings (i.e., single, continuously changing, and mixed). Hence, the overall error rates are increased compared to the single and continuously changing settings. In particular, the optimization-based method (i.e., TENT, SWR) shows an increased error rate from the single adaptation scenario (from 12.04 to 14.33 in TENT, from 10.26 to 13.24 in SWR when B=200) because it heavily depends on the current test batch statistics and updates the model parameter assuming that the update is helpful for the next batch prediction, which does not always hold in the mixed domain adaptation scenario. Therefore, adopting TTN to optimization-based method improves performance significantly.
>
> |Dataset   |Method  | 200 |  64 |  16 |  4  |  2  |  1  | Avg |
> |----------|--------|-----|-----|-----|-----|-----|-----|------
> CIFAR-10-C |**TTN** |12.16|	12.19|12.34|13.96|15.55|17.83|14.00
> -- |TENT    |14.33|	14.97|17.30|26.07|35.37|90.00|33.01
> -- |TENT+**TTN**|12.02|	12.04|12.20|13.77|15.42|16.40|13.64
> -- |SWR     |13.24|	13.06|16.57|26.08|38.65|91.03|59.54
> -- |SWR+**TTN**|11.89|	11.65|13.37|17.05|23.50|64.10|50.29
> CIFAR-100-C|**TTN**    |36.24|36.23|36.85|41.01|45.85|55.52|41.95
> --|TENT    |39.36|40.01|43.33|58.98|80.55|98.92|60.19
> --|TENT+**TTN**|36.29|36.23|36.89|41.38|46.65|57.95|42.57
> --|SWR     |37.84|37.93|44.37|59.50|78.66|98.95|33.10
> --|SWR+**TTN** |36.49|36.51|39.60|46.20|58.20|84.76|23.59
>
> **Q2.8: How to select the in Eqn. (6) to balance the two loss terms?**
>
> **A2.8**: We empirically set the MSE loss strength lambda as 1 through the ablation study (**Table 9**). Using the MSE regularizer prevents the learnable alpha from moving too far from the prior, thus letting alpha follow our intuition, i.e., putting smaller importance on the test batch statistics if the layer or channel is invariant to domain shifts. However, with too strong regularization (lambda=10.0), the overall error rates are high, which means the alpha needs to be sufficiently optimized. On the other hand, with too small regularization, the alpha may overfit to the training batch size (B=200) and lose generalizability to the smaller batch size.
>
> **Q2.9: It is quite interesting to see that the proposed method actually performs worse than non-learning-based normalization methods when the batch size is very small (2 or 1). Can the authors comment on that? Will a dynamic batch size in the post-training stage improve the performance in this case?**
>
> **A2.9**: We added variants of TTN in **Appendix B.3** and **Table 11**, which show stronger performance for small test batch sizes. Please refer to our responses [A1.2](https://openreview.net/forum?id=EQfeudmWLQ&noteId=at-ykx_Mktn) and [A1.3.2](https://openreview.net/forum?id=EQfeudmWLQ&noteId=1f_6NbEVz3l) for online and scaled TTN. As the reviewer pointed out, we observe that using the dynamic batch size in the post-training stage improves the performance for small test batch sizes (from 15.83% to 15.13% when B=2, and from 17.99% to 17.22% when B=1), while slightly deteriorating the performance for large test batch sizes (B=200, 64) in single and continual scenario. While optimizing the interpolating weight, we randomly sampled training batch size from the range of [4,200] for each iteration.

---

> > ### Author Response · Authors · 2022-11-17
> > **Summary of experimental results (A2.9)**
> >
> > **A2.9(continued)**: As we updated in **Table 11**, we added a variant of TTN, where alpha is post-trained with dynamically changing train batch size. The results are as follows:
> >
> > Method  |Eval Scenario| 200 |  64 |  16 |  4  |  2  |  1  | Avg |
> > --------|-------- |-----|-----|-----|-----|-----|-----|------
> > **TTN (offline, default)** | Single \& Cont. |11.67 |  11.80 | 12.13 | 13.93 |  15.83 |  17.99 | 13.89
> > **TTN (offline, default)** | Mixed |  12.16	| 12.19	| 12.34| 	13.96| 	15.55| 	17.83| 	14.00
> > TTN (dynamic train batch size) | Single \& Cont. |11.82	|12.04	|	12.17	|	13.60 	|	15.13 	|	17.22 	| 13.66
> > TTN (dynamic train batch size) | Mixed |12.12 |	12.01 |	11.91 | 13.43 | 14.76 | 17.23|13.58

---

> ### Author Response · Authors · 2022-11-16
> **Response to Reviewer 4Hgx (Q2.10-2.11)**
>
> **Q2.10: Why the normalization methods are not shown in Table 2 and Table 3? This looks like normalization methods cannot be used in these settings, but actually they can.**
>
> **A2.10**: The only difference between single (**Table 1**) and continuously changing (**Table 2**) settings is that the former resets the model parameters whenever the corruption type changes, while the latter continuously updates the model without resetting. Therefore, we omitted the norm methods results in Table 2 to avoid redundancy. We added the explanation in Section 3.2 and the table caption for clearance. We added norm method results in the mixed domain adaptation (**Table 3**) as we modified the setting (Please read the response [A.2.7](https://openreview.net/forum?id=EQfeudmWLQ&noteId=OEeD0m9pgS-) for the details).
>
> **Q2.11: While adding the proposed method improves upon the two optimization-based methods in Table 3, the results are even worse than applying the proposed method alone. This may weaken the proposed method and probably the authors can move it to the appendix.**
>
> **A2.11**: As we modified the mixed domain adaptation setting (please read our response to [A.2.7](https://openreview.net/forum?id=EQfeudmWLQ&noteId=OEeD0m9pgS-)), we updated the experimental result and observe that adding TTN to the optimization-based methods show better TTA performance than using our TTN method alone.

---

> > ### Comment · Reviewer_4Hgx · 2022-11-20
> > **Response to the authors**
> >
> > Thanks for providing the response together with additional details. Really appreciate the hard work!
> >
> > The response addresses most of my concerns. So I am willing to increase my score.
> >
> > Side note: Technical novelty could be very subjective so I do not use it as the major criterion. The important thing is whether the readers/community can learn any insights from the paper. And I believe this simple yet effective approach will inspire the following research in this test-time training/adaptation direction.

---

> ### Author Response · Authors · 2022-12-05
> **Thank you for raising the score!**
>
> We are deeply thankful for raising the score and providing valuable feedback with positive support.
>
> We will make thorough revisions to the final version of the manuscript and add all the details and insights you have provided.

---

### Official Review · Reviewer_yiJu · 2022-10-25

**Confidence:** 4
**Correctness:** 3
**Technical Novelty And Significance:** 2
**Empirical Novelty And Significance:** 2
**Recommendation:** 5

**Clarity, Quality, Novelty And Reproducibility:**

The paper has good clarity and enough details are provided to replicate the implementation. The novelty is marginal as this appears to be a fairly straightforward combination of two existing methods.

**Strength And Weaknesses:**

Strengths:

The technique is clear and fairly straightforward to implement.
Motivation describing pitfalls of CBN and TBN, and the need for better online test-time adaptation under flexible batch sizes are described well

Weaknesses:

It is unclear how the results would generalize across architectures. What will the results look like for transformers, which are currently ubiquitous in several applications? Will some tokens in a layer be adapted to the test set and the rest to the source? What would this mean in practice?

In Figure 2, why cannot alpha be learnt or adapted online? If it can be, then these methods should be compared with prompting (freezing certain layers will be analogous to using source-based normalization, while the prompting layers will be adapted to test set). It is unclear why an alpha learnt offline and frozen will continue to yield optimal results across a range of different test-time distributions.

The significance of results from Table 6 is not clear. These appear to be tested specifically on batch size 2 where TBN is  known to falter, but the results on only a few datasets appear significant. Similarly for the corruption benchmark, with a batch size of 1 or 2 (which is a fairly common real world scenario - data streaming and not evaluated in batches), TTN seems to be beat by one of the baselines. A better understanding and explanation of this is required. Is this because of optimizing the alphas offline?

Finally, what are the restrictions of how far the data distribution that the TTN model is optimized on and the real streaming test time data are? The results appear to indicate that there will be scenarios and assumptions mainly surrounding the dataset that alpha is optimized upon.


**Summary Of The Paper:**

The paper addresses a problem arising from the generality of source-based conventional batch normalization (CBN) or test-based batch normalization (TBN), with the former biasing the architecture to source distribution and the latter suffering from inaccuracies for small test batch sizes. The paper proposes combining CBN and TBN, and exposing a fraction of different layers in the architecture to TBN and CBN, thereby identifying those layers which need not be as sensitive to the test distribution. The results show promising performance of the proposed method, TTN, on the CIFAR and some DG benchmarks across several batch sizes.


**Summary Of The Review:**

Overall, I would rate the paper as below the acceptance threshold. While the method appears to show promise in certain settings, I would urge the authors to consider a wider selection of baseline architectures. Also, the results are not convincing enough about its use in a wider array of datasets or real-world scenarios (see the weakness section above).

---

> ### Author Response · Authors · 2022-11-16
> **Response to Reviewer yiJu (Q1.1, Q1.2)**
>
> We appreciate the thorough review and positive support. We hope our responses address all your concerns. Mainly, we added variants of TTN (e.g., online), which reveals further potential of our proposed method, and added more comparisons to show the significance of experimental results. Finally, we conducted a study on augmentation type and demonstrated the generalizability.
>
> **Q1.1: It is unclear how the results would generalize across architectures. What will the results look like for transformers, which are currently ubiquitous in several applications? Will some tokens in a layer be adapted to the test set and the rest to the source? What would this mean in practice?**
>
> **A1.1**: Currently, we proposed a method that calibrates batch normalization layers. Therefore, TTN is applicable to any backbone model that utilizes BN layers. It will be interesting to extend our approach to other normalization layers such as LayerNorm, and InstanceNorm, or different model architectures such as Transformers. We leave these as future work.
>
> **Q1.2: In Figure 2, why cannot alpha be learnt or adapted online? If it can be, then these methods should be compared with prompting (freezing certain layers will be analogous to using source-based normalization, while the prompting layers will be adapted to test set). It is unclear why an alpha learnt offline and frozen will continue to yield optimal results across a range of different test-time distributions.**
>
> **A.1.2**: Alpha can be learned during test time. We added results of the TTN online version (in **Appendix B.3** and **Table 11**), which further optimizes the post-trained alpha using the entropy minimization and mean-squared error loss between the updated alpha and the initial post-trained alpha. We set the learning rate as 1e-2, 2.5e-3, 5e-4, 1e-4, 5e-5, and 2.5e-5 by linearly decreasing according to the test batch size of 200, 64, 16, 4, 2, and 1 (Goyal et al., 2017). Adapting alpha online improved TTA performance compared to the original offline TTN in all three evaluation scenarios (single, continuously changing, and mixed).
> However, even without online adaptation, which requires extra computation in the test time, offline TTN already achieves promising results. We believe the post-trained alpha is already robust to a wide range of test-time distributions because we optimized the alpha according to the domain-shift sensitivity using various augmentation types. We additionally conducted an ablation study on augmentation type to show the generalizability of the offline TTN  (**Appendix B.4** and **Table 5**).
>
> [1] Priya Goyal, Piotr Dollár, Ross Girshick, Pieter Noordhuis, Lukasz Wesolowski, Aapo Kyrola, Andrew Tulloch, Yangqing Jia, and Kaiming He. Accurate, large minibatch sgd: training imagenet in 1 hour. arXiv preprint arXiv:1706.02677, 2017.

---

> > ### Author Response · Authors · 2022-11-17
> > **Summary of experimental results (A1.2)**
> >
> > **A1.2(continued)**: As we updated in **Table 11**, we added TTN online version and the results are as follows:
> >
> > Method  |Eval Scenario| 200 |  64 |  16 |  4  |  2  |  1  | Avg |
> > --------|-------- |-----|-----|-----|-----|-----|-----|------
> > **TTN (offline, default)** | Single \& Cont. |11.67 |  11.80 | 12.13 | 13.93 |  15.83 |  17.99 | 13.89
> > **TTN (offline, default)** | Mixed |  12.16	| 12.19	| 12.34| 	13.96| 	15.55| 	17.83| 	14.00
> > TTN (online) | Single |11.39|	11.64|	11.97|	13.70|	15.41|	17.49|	13.60
> > TTN (online) | Cont. | 11.67|	11.96|	12.15|	13.90|	15.67|	17.72|	13.85
> > TTN (online) | Mixed | 12.04|	12.04|	12.06 |	13.90|	15.46|	17.62|	13.85

---

> ### Author Response · Authors · 2022-11-16
> **Response to Reviewer yiJu (Q1.3, Q1.4)**
>
> **Q1.3.1: The significance of results from Table 6 is not clear. These appear to be tested specifically on batch size 2 where TBN is known to falter, but the results on only a few datasets appear significant.**
>
> **A1.3.1**: To clarify the significance of segmentation results, we added more comparisons in **Table 6** (please refer to our response [A2.5](https://openreview.net/forum?id=EQfeudmWLQ&noteId=Ny2Io7namGq). The results show that even when exploiting test batch statistics for standardization in BN layers (TBN) or updating the model parameter on top of TBN (TENT, SWR) does not improve the model performance (i.e., perform worse than the source model), adopting TTN helps the model make good use of the strength of test batch statistics.
>
> Moreover, considering that the backbone network is pre-trained using a training batch size of 8 (Chen et. al, 2018, Choi et al, 2021), we believe that using a test batch size of 2 in semantic segmentation is not as harsh as in image classification tasks. We additionally evaluated TBN methods using larger test batch sizes (B=4, 8) and observed that the performance gap along different test batch sizes is not too significant. Importantly, we observe that TTN with B=2 beats TBN B=4 and B=8.
>
> * Please refer to the updated table [here](https://openreview.net/forum?id=EQfeudmWLQ&noteId=JYZ2g15axQ).
> |  Method | BDD-100K  | Mapiliary  | GTAV  | SYNTHIA  | Cityscapes
> |---|---|---|---|---|---|
> | TBN (B=2) | 43.12 | 47.61 | 42.51 | 25.71 |  72.94
> | TBN (B=4) | 45.64   | 49.17  | 44.26  | 25.96  | 74.29
> | TBN (B=8) | 46.42   | 49.38  | 44.81  | 25.97  | 75.42
> | **Ours (TTN) (B=2)** | 47.40 |56.92| 44.71| 26.68 |75.09
>
>
> [1] Chen, L. C., Zhu, Y., Papandreou, G., Schroff, F., & Adam, H. (2018). Encoder-decoder with atrous separable convolution for semantic image segmentation. In Proceedings of the European conference on computer vision (ECCV) (pp. 801-818).
>
> [2] Choi, S., Jung, S., Yun, H., Kim, J. T., Kim, S., & Choo, J. (2021). Robustnet: Improving domain generalization in urban-scene segmentation via instance selective whitening. In Proceedings of the IEEE/CVF Conference on Computer Vision and Pattern Recognition (pp. 11580-11590).
>
> **Q1.3.2:  Similarly for the corruption benchmark, with a batch size of 1 or 2 (which is a fairly common real world scenario - data streaming and not evaluated in batches), TTN seems to be beat by one of the baselines. A better understanding and explanation of this is required. Is this because of optimizing the alphas offline?**
>
> **A1.3.2**: We added variants of TTN in **Appendix B.3** and **Table 11**, which show stronger performance for small test batch sizes. First, the online version keeps optimizing the interpolating weight in test time as mentioned in our response [A1.2](https://openreview.net/forum?id=EQfeudmWLQ&noteId=at-ykx_Mktn). Secondly, we scaled down the alpha value by multiplying a constant ranging from 0 to 1. Finally, we dynamically changed the training batch size in the post-training stage (please refer to our response [A2.9](https://openreview.net/forum?id=EQfeudmWLQ&noteId=Ny2Io7namGq) for the last variant).
>
> The intuition for the scaled TTN version comes from the results of normalization-based methods where we set the alpha as a constant value ranging from 0 to 1 (**Table 10**). Alpha = 0 is identical to CBN (Ioffe & Szegedy, 2015), and alpha = 1 is identical to TBN (Nado et al., 2020). We observe that the lower alpha, i.e., putting less importance on test batch statistics, shows better performance for small test batch sizes. This observation is analogous to the finding of the previous work (Schenider et al., 2020), which suggested alpha as 0.44, 0.33, 0.2, 0.11, 0.06, and 0.06 for test batch size 200, 64, 16, 4, 2, and 1. Following the findings, we lowered the interpolating weight by multiplying a constant scale value, ranging (0,1), to the optimized TTN alpha. Here, we empirically set the scale value as 0.4. We believe that a strategy for adjusting the interpolating weight in accordance with the test batch size should be further explored in future work.
>
> **Q1.4: Finally, what are the restrictions of how far the data distribution that the TTN model is optimized on and the real streaming test time data are? The results appear to indicate that there will be scenarios and assumptions mainly surrounding the dataset that alpha is optimized upon.**
>
> **A1.4**: To show the robustness of our method in terms of the domain gap between augmentation type and test data distributions, we conducted an ablation study. We observe that TTN shows robust and domain-invariant results. Please refer to our response [A4.1, A4.2, A4.3](https://openreview.net/forum?id=EQfeudmWLQ&noteId=EJvV_CM-F_T) and revised results in **Appendix B.4** and **Figure 5**.

---

> > ### Author Response · Authors · 2022-11-17
> > **Summary of experimental results (A1.3.2)**
> >
> > **A1.3.2(continued)**: As we updated in **Table 11**, we added variants of TTN, which show stronger performance when test batch size is small. The results are as follows:
> >
> > Method  |Eval Scenario| 200 |  64 |  16 |  4  |  2  |  1  | Avg |
> > --------|-------- |-----|-----|-----|-----|-----|-----|------
> > **TTN (offline, default)** | Single \& Cont. |11.67 |  11.80 | 12.13 | 13.93 |  15.83 |  17.99 | 13.89
> > **TTN (offline, default)** | Mixed |  12.16	| 12.19	| 12.34| 	13.96| 	15.55| 	17.83| 	14.00
> > TTN (scaled) | Single \& Cont. | 13.20  |	13.38 |	13.35 |	13.88 |	14.54|	15.17|	13.92
> > TTN (scaled) | Mixed | 13.17 | 13.05 |	13.17 |	 13.74 |	14.36|	15.09|	13.76
> >
> > Moreover, we added simple baselines, where we set alpha as a constant value in **Table 10**. Inspired by this experiment result, we added scaled TTN as a variant. The results are as follows:
> >
> > Constant alpha  | 200 |  64 |  16 |  4  |  2  |  1  | Avg |
> > |-------- |-----|-----|-----|-----|-----|-----|------
> > 0 (CBN)	|18.26|	18.39|	18.26|	18.26|	18.25|	18.25|	18.28|
> > 0.1	|13.95|	14.1|	14.05|	14.65|	15.14|	15.45|	14.56|
> > 0.2	|12.46|	12.66|	12.89|	14.3|	15.53|	15.64|	13.91|
> > 0.3	|12.05|	12.29|	12.72|	15.18|	17.35|	17.42|	14.50|
> > 0.4	|12.13|	12.41|	13.12|	16.69|	19.81|	20.51|	15.78|
> > 0.5	|12.42|	12.78|	13.73|	18.32|	22.52|	24.88|	17.44|
> > 0.6	|12.88|	13.32|	14.48|	20.02|	25.17|	31.97|	19.64|
> > 0.7	|13.37|	13.9|	15.23|	21.75|	27.91|	46.65|	23.14|
> > 0.8	|13.82|	14.37|	15.94|	23.44|	30.59|	77.15|	29.22|
> > 0.9	|14.18|	14.8|	16.58|	24.94|	33.12|	89.81|	32.24|
> > 1 (TBN)	|14.5|	15.15|	17.1|	26.29|	35.67|	90	|33.12|

---

> > > ### Comment · Reviewer_yiJu · 2022-12-06
> > > **response**
> > >
> > > We appreciate the response by the authors. While some of my concerns have been addressed, a few still remain. Since the idea behind the paper is a fairly straightforward combination of two existing methods, I feel the strength of the paper should be in the results. I am still not convinced by the significance angle even with the additional experiments. TTN with B=2 is compatible and sometimes inferior to TBN (B=8). Confidence intervals are not provided which makes it difficult to assess what a 1% difference would mean. If this can be answered, I can raise the rating to a 6.

---

> > > > ### Author Response · Authors · 2022-12-07
> > > > **Thank you for your response! & Answer for the further question**
> > > >
> > > > Thank you very much for your further question and reconsideration of the score.
> > > >
> > > > We have added the results of TTN with B=4 and 8 for more comparisons. The results demonstrate that TTN is robust to the test batch sizes. In other words, the performance difference across the test batch size is small when using TTN (TTN with B=2, 4, and 8). The results are averaged over 3 runs (i.e., using 3 independently optimized alpha), and the standard deviation is denoted with a $\pm$ sign. We omitted the standard deviation for TBN, which is 0.0 for every result since no optimization is required.
> > > >
> > > > As can be seen in other experiments (e.g., Figure 1, Table 1, and Table 3), TBN improves the model performance under distribution shift, when the test batch size is large enough and the estimated statistics are reliable (Nado et al., 2020, Schneider et al., 2020). Since the backbone network we used for the semantic segmentation experiments is trained with a train batch size of 8, we assume that test batch statistics (i.e., mean and variance for BN standardization) estimated from the test batch with B=8 are sufficiently reliable. Accordingly, TBN with B=8 shows compatible results. However, when B becomes small (i.e., in a more practical scenario), problematic test batch statistics are estimated, and thus TBN suffers from the performance drop (the average accuracy for TBN 48.40% (B=8) vs. 46.38% (B=2)) while TTN keeps showing the robust performance (the average accuracy for TTN 50.39% (B=8) vs. 50.16% (B=2)). It is worth noting that TTN outperforms TBN by 3.77% in average accuracy when B=2, i.e., in the most practical evaluation setting, and by 2.54% and 1.99% for B=4 and 8, respectively.
> > > >
> > > >
> > > >
> > > > | Method	| BDD-100K	| Mapiliary	| GTAV	| SYNTHIA	| Cityscapes	| avg
> > > > |---|---|---|---|---|---|---|
> > > > |TBN (B=2)          |	43.12|	47.61	|42.51|	25.71|	72.94|	46.38
> > > > |**Ours (TTN) (B=2)**   |	**47.40**($\pm$ 0.02)|	**56.88**($\pm$ 0.04)|	**44.69**($\pm$ 0.03)|	**26.68**($\pm$ 0.01)|	**75.09**($\pm$ 0.01)|**50.15**($\pm$ 0.01)
> > > > |TBN (B=4)          |	45.64 |	49.17	|44.26|	25.96|	74.29|	47.86
> > > > |**Ours (TTN) (B=4)**   |	**47.72**($\pm$ 0.01)|	**57.11**($\pm$ 0.01)|	**45.08**($\pm$ 0.02)|	**26.52**($\pm$ 0.01)|	**75.56**($\pm$ 0.01)|**50.40**($\pm$ 0.01)
> > > > |TBN (B=8)          |	46.42 |	49.38	|44.81|	25.97|	75.42|	48.40
> > > > |**Ours (TTN) (B=8)**   |	**47.25**($\pm$ 0.02) |	**57.28**($\pm$ 0.02)|	**45.13**($\pm$ 0.03)|	**26.45**($\pm$ 0.01)|	**75.82**($\pm$ 0.01)|	**50.39**($\pm$ 0.01)

---

> > > > ### Author Response · Authors · 2022-12-10
> > > > **A gentle reminder to Reviewer yiJu**
> > > >
> > > > Dear reviewer yiJu,
> > > >
> > > > Thank you again for spending time reviewing our paper. Since the second discussion stage will end in 2 days, we give a friendly reminder of our response to your further question.
> > > >
> > > > We are delighted that our responses have addressed your insightful concerns, which further enhanced the quality of our paper. We would really appreciate it if you could consider our response. If there are any following comments or questions, please kindly let us know.
> > > >
> > > > Best,
> > > >
> > > > Paper 4623 authors

---

> ### Author Response · Authors · 2022-12-05
> **Kind reminder for discussion**
>
> Dear reviewer yiJu,
>
> We deeply appreciate your valuable feedback on improving our paper.
>
> We have already provided detailed responses to the comments, which can be summarized as follows:
>
> - In this paper, we have focused on model architectures utilizing batch normalization layers. We leave expanding TTN to other architectures, such as transformers, as future work. We will revise the final version of our manuscript by adding this discussion in the limitation section to respond to generalizability across architectures (Q1.1). On the other hand, to demonstrate the generalizability of our method, we have conducted experiments on the semantic segmentation task (Table 6), showing significant performance improvements not only when using TTN as alone but also when utilizing TTN in combination with backpropagation-based approaches. We believe that the results of TTN in different tasks (i.e., image classification and semantic segmentation) demonstrate the wide applicability of TTN.
> - As a response to the suggestion of TTN online version (Q1.2), we have added the results of online TTN and opened the possibility of further improvement, while the offline version is still convincing.
> - Moreover, we have added more comparisons for the semantic segmentation experiment (Q1.3.1) to show the significance of the results and added TTN variants (Q1.3.2) which show strong performance for small test batch sizes.
> - Finally, as a response to the concern about the robustness of TTN in terms of the choice of augmentation type (Q1.4), we have conducted extensive augmentation ablations and provided important insight by showing amazing results (ref. [Response from Reviewer RNFt](https://openreview.net/forum?id=EQfeudmWLQ&noteId=yNEfnMMo1s)).
>
> We are looking forward to receiving any further comments or suggestions. We will do our best to respond to them.
>
> Best,
>
> Paper 4623 authors

---

### Author Response · Authors · 2022-11-16
**Summary of Revision**

**We appreciate all four reviewers for their constructive feedback and valuable comments.**

**We found our strengths from the reviews as follows:**
- The proposed approach is technically sound, straightforward to implement, effective and well-motivated, tackling an interesting and important task (handling plenty of test-time scenarios).
- The paper is well-written, providing comprehensive discussion and detailed ablation studies which help the readers to better understand the method.
- The experiments show empirically successful performance in various evaluation settings.

**We have updated the manuscript after thorough revisions. A summary of the main changes are as follows:**
- We analyzed the generalizability of our method through conducting a study on augmentation type in Appendix B.4 and Figure 5. (Reviewer yiJu, RNFt)
- We added more comparisons in image classification and semantic segmentation experiments. Table 1, 2, and 6 are updated. Additionally, we added simple baseline, where we set various constant value as the inperpolating weight in Appendix B.2. (Reviewer yiJu, 4Hgx)
- We added variants of our proposed method (online, scaled, and dynamic training batch size) and showed stronger result with small test batch sizes (Appendix B.3 and Table 11). (Reviewer yiJu, 4Hgx)
- We updated mixed domain evaluation setting and the experimental results in Table 3. (Reviewer 4Hgx)
- We thoroughly revised the manuscript and added missing references and details (TBN version). (Reviewer 4Hgx)
- We conducted ablation study on regularization loss term in Table 9. (Reviewer 4Hgx)
- We added PyTorch-friendly pseudo code and implementation details in Appendix A.3 and A.1 to improve the reproducibility. (Reviewer EgJB)

**Please refer to the revised manuscript to see the details and complete experimental results.**
**We hope our revisions and responses address all reviewer’s concerns, and any additional comments and clarifications are greatly appreciated.**

---

### Decision · Program_Chairs · 2023-01-20

**Decision:**

Accept: poster

**Justification For Why Not Higher Score:**

 I think the paper is still somewhat borderline, and several of the new results that are most compelling were only mentioned in the discussion (and thus likely were somewhat add-on experiments done after the original submission).  The core idea also seems to be a nice but still incremental method in a long line of test time adaptation approaches, which is great to include but likely not sufficient for a spotlight.

**Justification For Why Not Lower Score:**

 While there were some negative scores for this paper, I think the additional experiments and overall simplicity and performance of the method make a very strong case for inclusion here.

**Metareview: Summary, Strengths And Weaknesses:**

Thank you for your submission to ICLR.  The paper presents a new approach to normalization that interpolates between traditional batch normalization (using average statistics at test time) with test-time-applied batch normalization (which requires entire batches of data at test time, thus referred to as the tranductive batch norm, TBN, in this paper).

There was some disagreement about this paper from the reviewers, but ultimately I am more persuaded by the positive perspectives on this paper.  In particular, I think that the benefits and relative simplicity of the approach are compelling, and that most of the unaddressed negative comments were fairly generic (regarding why not evaluating on different architectures, etc), that may or may not really be relevant here (it's unclear, e.g., if the approach really needs to consider something that applies to architectures that don't contain BN layers).

The authors presented a vast number of experiments in response to reviewer concerns, including some fairly impressive results on larger-scale problems.  While these presentations border on presenting new results (typically not the goal of the rebuttal period), my general feeling is that they still represent compelling evidence for the method, and shouldn't be dismissed.  Overall, my main recommendation, then, is that the authors synthesize these many new results into the paper, perhaps even changing some of the focus given these rather impressive new results.

**Note From Pc:**

if the above contains the word "oral" or "spotlight" please see: "oral" presentation means -> notable-top-5% and "spotlight" means -> notable-top-25%. As stated in our emails, we are disassociating presentation type from AC recommendations